# Optimistic Dual Extrapolation for Coherent Non-monotone Variational Inequalities

Chaobing Song[†][*]    Zhengyuan Zhou[+]    Yichao Zhou[‡]    Yong Jiang[†]    Yi Ma[‡]

[†]Tsinghua-Berkeley Shenzhen Institute, Tsinghua University
songcb16@mails.tsinghua.edu.cn,   jiangy@sz.tsinghua.edu.cn
[‡]Department of EECS, University of California, Berkeley
zyc@berkeley.edu, yima@eecs.berkeley.edu
[+]Stern School of Business, New York University, zzhou@stern.nyu.edu

## Abstract

The optimization problems associated with training generative adversarial neural networks can be largely reduced to certain *non-monotone* variational inequality problems (VIPs), whereas existing convergence results are mostly based on monotone or strongly monotone assumptions. In this paper, we propose *optimistic dual extrapolation (OptDE)*, a method that only performs *one* gradient evaluation per iteration. We show that OptDE is provably convergent to *a strong solution* under different coherent non-monotone assumptions. In particular, when a *weak solution* exists, the convergence rate of our method is $O(1/\epsilon^2)$, which matches the best existing result of the methods with two gradient evaluations. Further, when a *$\sigma$-weak solution* exists, the convergence guarantee is improved to the linear rate $O(\log \frac{1}{\epsilon})$. Along the way–as a byproduct of our inquiries into non-monotone variational inequalities–we provide the near-optimal $O\left(\frac{1}{\epsilon} \log \frac{1}{\epsilon}\right)$ convergence guarantee in terms of restricted strong merit function for monotone variational inequalities. We also show how our results can be naturally generalized to the stochastic setting, and obtain corresponding new convergence results. Taken together, our results contribute to the broad landscape of variational inequality–both non-monotone and monotone alike–by providing a novel and more practical algorithm with the state-of-the-art convergence guarantees.

## 1   Introduction

Variational inequality (VI) provides a principled framework for minimax problems via their first-order optimality conditions. Given a closed convex set $\mathcal{W} \subset \mathbb{R}^d$ and an operator $F : \mathcal{W} \to \mathbb{R}^d$, the variational inequality problem VIP$(F, \mathcal{W})$ aims to find a solution $\boldsymbol{w}^* \in \mathcal{W}$ such that:

$$\forall \boldsymbol{w} \in \mathcal{W}, \ \langle F(\boldsymbol{w}^*), \boldsymbol{w} - \boldsymbol{w}^* \rangle \geq 0, \tag{1}$$

where $\boldsymbol{w}^*$ is called a *strong solution* of VIP$(F, \mathcal{W})$. For the minimax problem

$$\min_{\boldsymbol{x} \in \mathcal{X}} \max_{\boldsymbol{y} \in \mathcal{Y}} f(\boldsymbol{x}, \boldsymbol{y}), \tag{2}$$

let $\mathcal{W} \equiv \mathcal{X} \times \mathcal{Y}, \boldsymbol{w} \equiv \left[\begin{smallmatrix} \boldsymbol{x} \\ \boldsymbol{y} \end{smallmatrix}\right], F(\boldsymbol{w}) \equiv \left[\begin{smallmatrix} \nabla_{\boldsymbol{x}} f(\boldsymbol{x}, \boldsymbol{y}) \\ -\nabla_{\boldsymbol{y}} f(\boldsymbol{x}, \boldsymbol{y}) \end{smallmatrix}\right]$. Then solving (1) is equivalent to finding a first-order Nash equilibrium of the minimax problem (2) [32].

---

[*]This work was conducted during Chaobing Song's visit to Professor Yi Ma's group at UC Berkeley.

**Convex-Concave Minimax Problems.** The operator $F(\boldsymbol{w})$ will be *monotone* if

$$\forall \boldsymbol{w}, \boldsymbol{v} \in \mathcal{W}, \ \langle F(\boldsymbol{w}) - F(\boldsymbol{v}), \boldsymbol{w} - \boldsymbol{v} \rangle \geq 0. \tag{3}$$

VI with monotone operators has been well studied, which provides a concise and optimal framework for convex-concave minimax problems [29]. For monotone VIP$(F, \mathcal{W})$, it is well known that the strong solution satisfying (1) is also equivalent to the solution $\boldsymbol{w}^* \in \mathcal{W}$ satisfying:

$$\forall \boldsymbol{w} \in \mathcal{W}, \ \langle F(\boldsymbol{w}), \boldsymbol{w} - \boldsymbol{w}^* \rangle \geq 0, \tag{4}$$

where $\boldsymbol{w}^*$ is called a *weak solution* of VIP$(F, \mathcal{W})$. A classical result [29] under the monotone and Lipschitz continuous assumptions is that the *Mirror-Prox* algorithm [29] can converge to an $\epsilon$-accurate weak solution in terms of ergodic averaging in $O(1/\epsilon)$ iterations, which is optimal for first-order methods in solving monotone VIPs [30, 33]. Nemirovski's Mirror-Prox is a non-Euclidean extension of the extragradient method [22] from the perspective of mirror descent. Another important non-Euclidean extension is Nesterov's *dual extrapolation* [31] from the perspective of dual averaging, which also has the optimal $O(1/\epsilon)$ convergence rate. The main difference between mirror descent and dual averaging is the way of combining the constraint (or the regularization term if exists) into the projection (or the proximal) step [31].

Despite obtaining the optimal convergence rate, both Mirror-Prox and dual extrapolation are *two-call* extragradient methods that need to evaluate gradients *twice per iteration*. In some contexts such as training deep neural networks, evaluating gradients can be expensive. Thus it will have significant practical benefits if we only need one gradient evaluation per iteration and still maintain the same convergence rate. In terms of *single-call* methods for minimax problems, vanilla gradient descent ascent (and its mirror descent generalizations) might be a natural choice. Unfortunately, it is not guaranteed and it can diverge even in simple monotone settings [24]. Consequently, after the (two-call) extragradient method [22], several *single-call* extragradient methods [35, 3, 6, 27] have been analyzed under the monotone setting and share the same convergence rates with Mirror-Prox and dual extrapolation [17]. However, there is an increasing trend in applying these single-call extragradient methods to stabilize the training of generative adversarial networks (GAN) [8, 12, 34], which is *nonconvex-nonconcave* in general and hence has remained underexplored.

**Nonconvex-Nonconcave Minimax Problems.** Despite the well-developed convergence theory for monotone VIPs and thus for convex-concave minimax problems, many minimax problems arising in modern machine learning are nevertheless *nonconvex-nonconcave*, such as GAN [14], adversarial training [15], gradient reversal for domain adaption [11], and multi-agent reinforcement learning [38]. As a result, the corresponding VI is not monotone and the aforementioned theoretical guarantees for monotone VIPs no longer apply. First, for non-monotone VIPs, it is nontrivial to obtain the rate of convergence to a weak solution, thus one may explore the rate of convergence to a strong solution instead. Second, without the monotone property, the ergodic averaging technique [22] will no longer have theoretical guarantees, thus we might need to choose the *last iterate* or *best iterate*. However, the classical convergence result [29] said little about the rate of convergence to a weak solution or the convergence of last iterate or best iterate.[2]

To obtain theoretical guarantees beyond the monotone setting, a common approach is to relax the lower bound (3) in the monotone assumption. Along this research line, several more general assumptions have been proposed, such as the *pseudo-monotone* assumption [20, 16] and its variants [19], and the *generalized monotone* assumption [7]. In the machine learning community, similar concepts have also been proposed, such as variational coherence [41, 42]. For simplicity, we coin the problem class along this research line as *coherent non-monotone variational inequalities*. Among them, [7] is the first to provide explicit global convergence results such that the best iterate of the N-EG method [7] can converge to an $\epsilon$-accurate strong solution in $O(1/\epsilon^2)$ iterations under the generalized monotone and Lipschitz continuous assumptions. However, N-EG needs to evaluate gradient *twice per iteration*, which is less desirable when gradient evaluation is expensive. For the single-call extragradient method [4], under a second-order condition[3], very recently [17] has provided local linear convergence results in certain non-monotone setting, while the constants in these results remain implicit. The following problem remains open: *Can single-call extragradient methods have explicit global convergence results beyond the monotone setting?*

Table 1: **Iteration complexity for finding an $\epsilon$-accurate solution in the deterministic setting.** (In both Tables 1 and 2, "—" denotes the corresponding results are not known or can not be obtained.)

| Convergence measure | Merit function (Definition 1) | | Distance $\|\cdot - \boldsymbol{w}^*\|^2$ |
|---|---|---|---|
| Algorithm | N-EG [7] | OptDE (this Paper) | OptDE (this Paper) |
| Weak solution exists | $O(1/\epsilon^2)$ | $O(1/\epsilon^2)$ | — |
| $\sigma$-weak solution exists | — | $O(\log \frac{1}{\epsilon})$ | $O(\log \frac{1}{\epsilon})$ |
| No. of gradient calls | 2 | 1 | 1 |

Table 2: **Stochastic oracle complexity for finding an expected $\epsilon$-accurate solution in the stochastic setting.**

| Convergence measure | Merit function (Definition 1) | | Distance $\mathbb{E}[\|\cdot - \boldsymbol{w}^*\|^2]$ | |
|---|---|---|---|---|
| Algorithm | SEG [18] | SOptDE (this paper) | ESA [19] | SOptDE (this paper) |
| Weak solution exists | $O(1/\epsilon^4)$ | $O(1/\epsilon^4)$ | — | — |
| $\sigma$-weak solution exists | — | $O(1/\epsilon^2 \log \frac{1}{\epsilon})$ | $O(1/\epsilon)$ | $O(1/\epsilon)$ |
| No. of gradient calls | 2 | 1 | 2 | 1 |

**Contributions of This Paper.** In this paper we develop an *Optimistic Dual Extrapolation (OptDE)* method that provably converges to a strong solution for coherent non-monotone VIPs. The OptDE method can be viewed as a single-call variant of Nesterov's dual extrapolation that maintains its "anticipatory" properties. We characterize convergence rates of the best iterate[4] of OptDE under two coherent non-monotone assumptions, where the merit function is given in Definition 1 and $\|\cdot\|$ is the natural norm used in algorithms. As shown in Table 1, when the problem has a *weak solution* $\boldsymbol{w}^*$, our method matches the best known rate $O(1/\epsilon^2)$ of N-EG [7]. Further strengthening the assumption to that a *$\sigma$-weak solution* $\boldsymbol{w}^*$ exists with $\sigma > 0$ – nevertheless a weaker condition than the strongly monotone assumption required in previous work, we are able to obtain a linear convergence rate of $O(\log \frac{1}{\epsilon})$. For this setting, we can also use the distance $\|\cdot - \boldsymbol{w}^*\|^2$ to measure the progress and obtain a linear convergence result; meanwhile, despite not shown in Table 1, we also obtain a linear convergence result of the last iterate. Our result shows that even under the two coherent non-monotone assumptions, the convergence rate of single-call extragradient methods can be comparable to that of the N-EG method with two gradient evaluations per iteration.

Our coherent non-monotone analysis for the setting that a $\sigma$-weak solution exists has two meaningful corollaries about best iterate and last iterate *in the monotone setting*, respectively: With *a regularization trick*, both the best iterate and last iterate[5] of OptDE can be an $\epsilon$-accurate solution in $O(\frac{1}{\epsilon} \log \frac{1}{\epsilon})$ number of iterations. To our knowledge, the near-optimal result $O(\frac{1}{\epsilon} \log \frac{1}{\epsilon})$ for attaining an $\epsilon$-accurate strong solution was only appeared in [9] very recently with a two-loop Halpern iteration method, while our result is obtained by the simpler single-loop single-call OptDE method.

Meanwhile, we extend the OptDE algorithm to the stochastic setting as *Stochastic OptDE (SOptDE)* and show that our results in the deterministic setting can be naturally generalized to the *stochastic setting*. This allows us to characterize the stochastic oracle complexity (*i.e.,* the number of stochastic oracles we access) of SOptDE under the coherent non-monotone assumptions. The results under the stochastic setting are summarized in Table 2.[6] As we see, the results match the best-known results of SEG [7] [18] and ESA [19] respectively, while both SEG and ESA need two gradient evaluations per iteration. Meanwhile, under the assumption that a $\sigma$-weak solution exists, we obtain the first theoretical guarantee in terms of the merit function in Definition 1.

Last but not least, different from N-EG [7] and ESA [19], the proposed OptDE and SOptDE algorithms only need the norm square $\|\cdot\|^2$ being strongly convex but not necessarily globally Lipschitz continuous, which will be significant if $\|\cdot\|$ is a non-Euclidean norm: $\|\cdot\|^2$ can not be strongly convex and globally Lipschitz continuous simultaneously in general.

## 2  Technical Assumptions

**Notations:** For $K \in \mathbb{Z}_+$, let $[K] := \{1, 2, \ldots, K\}$. Let lower case boldface alphabets denote vectors, such as $\boldsymbol{x} \in \mathbb{R}^d$ and lower case alphabets with subscript denote elements, such as $x_1, x_2, \ldots, x_d$. Let $\| \cdot \|$ denote a general norm. Let $\| \cdot \|_*$ denote the dual norm of $\| \cdot \|$ defined by $\|\boldsymbol{y}\|_* := \max_{\|\boldsymbol{x}\| \leq 1} \langle \boldsymbol{x}, \boldsymbol{y} \rangle$. For $\boldsymbol{x} \in \mathbb{R}^d$ and $p \geq 1$, let $\|\boldsymbol{x}\|_p := \left( \sum_{i=1}^d |x_i|^p \right)^{\frac{1}{p}}$.

To measure the accuracy of iterates to a strong solution, we consider the following "restricted strong merit function".

**Definition 1 (Restricted strong merit function)** *$\tilde{w} \in \mathcal{W}$ is an $\epsilon$-accurate strong solution of the VIP$(F, \mathcal{W})$ with a fixed parameter $D > 0$ if*

$$\sup_{\boldsymbol{w} \in \mathcal{W}, \|\boldsymbol{w} - \tilde{\boldsymbol{w}}\|_2 \leq D} \langle F(\tilde{\boldsymbol{w}}), \tilde{\boldsymbol{w}} - \boldsymbol{w} \rangle \leq \epsilon. \tag{5}$$

With $\epsilon \to 0$ and $D \to +\infty$, Definition 1 becomes the definition of the strong solution in (1). In the nonconvex-nonconcave minimax setting, Definition 1 has been proposed as the definition of the $\epsilon$-accurate first-order Nash equilibrium [32]. If $\mathcal{W}$ is a bounded set, then we still have an effective measure even if $D \to +\infty$; if $\mathcal{W}$ is unbounded, then $D$ needs to be a finite positive parameter. To give a unified measure for both bounded and unbounded settings, we set $D$ to be a finite positive parameter.

Throughout this paper, we make the following standard Lipschitz continuous assumption.

**Assumption 1** *For the VIP$(F, \mathcal{W})$ in (1), $\forall \boldsymbol{w}, \boldsymbol{v} \in \mathcal{W}$, $\|F(\boldsymbol{w}) - F(\boldsymbol{v})\|_* \leq L\|\boldsymbol{w} - \boldsymbol{v}\|$, where $L > 0$ is the Lipschitz constant.*

Meanwhile, we assume that the (possible non-Euclidean) norm $\| \cdot \|$ satisfies Assumption 2.

**Assumption 2** *$\frac{1}{2}\|\boldsymbol{w}\|^2$ is $\gamma$-strongly convex ($0 < \gamma \leq 1$) with respect to (w.r.t.) $\| \cdot \|$ and the dual norm of gradient $\nabla \frac{1}{2}\|\boldsymbol{w}\|^2$ is bounded by $\delta\|\boldsymbol{w}\| (\delta > 0)$:*

$$\frac{1}{2}\|\boldsymbol{w}\|^2 \geq \frac{1}{2}\|\boldsymbol{v}\|^2 + \langle \nabla \frac{1}{2}\|\boldsymbol{v}\|^2, \boldsymbol{w} - \boldsymbol{v} \rangle + \frac{\gamma}{2}\|\boldsymbol{w} - \boldsymbol{v}\|^2, \tag{6}$$

$$\left\| \nabla \frac{1}{2}\|\boldsymbol{w}\|^2 \right\|_* \leq \delta\|\boldsymbol{w}\|. \tag{7}$$

From [1], $\frac{1}{2}\| \cdot \|_p^2 (1 < p \leq 2)$ is $(p-1)$-strongly convex w.r.t. $\| \cdot \|_p$. Without loss of generality, in Assumption 2, we assume $0 < \gamma \leq 1$. For all the norm setting $\frac{1}{2}\| \cdot \|_p^2 (1 < p \leq 2)$, we have $\delta = 1$.

For the norm $\| \cdot \|$, we define the prox-mapping as

$$P_{\boldsymbol{v}}(\boldsymbol{w}) := \arg\min_{\boldsymbol{z} \in \mathcal{W}} \left\{ \langle \boldsymbol{w}, \boldsymbol{z} \rangle + \frac{1}{2\gamma}\|\boldsymbol{z} - \boldsymbol{v}\|^2 \right\}, \tag{8}$$

and assume that it can be solved efficiently. Meanwhile, we also define the corresponding Bregman divergence of $\frac{1}{2}\| \cdot \|^2$: $\forall \boldsymbol{w}, \boldsymbol{v} \in \mathcal{W}$,

$$V_{\boldsymbol{v}}(\boldsymbol{w}) := \frac{1}{2}\|\boldsymbol{w}\|^2 - \frac{1}{2}\|\boldsymbol{v}\|^2 - \langle \nabla \frac{1}{2}\|\boldsymbol{v}\|^2, \boldsymbol{w} - \boldsymbol{v} \rangle. \tag{9}$$

Obviously we have $V_{\boldsymbol{v}}(\boldsymbol{w}) \geq \frac{\gamma}{2}\|\boldsymbol{w} - \boldsymbol{v}\|^2$.

Then we make Assumptions 3 and 4 for the coherent non-monotone VIP$(F, \mathcal{W})$ we study.

**Assumption 3 (Existence of a weak solution)** *For the VIP$(F, \mathcal{W})$ in (1), there exists a weak solution $\boldsymbol{w}^* \in \mathcal{W}$ such that $\forall \boldsymbol{w} \in \mathcal{W}, \langle F(\boldsymbol{w}), \boldsymbol{w} - \boldsymbol{w}^* \rangle \geq 0$.*

**Assumption 4 (Existence of a $\sigma$-weak solution)** *For the VIP$(F, \mathcal{W})$ in (1), given $\boldsymbol{w}_0 \in \mathcal{W}$, there exists a $\sigma$-weak solution $\boldsymbol{w}^* \in \mathcal{W}$ with parameter $\sigma > 0$ such that $\forall \boldsymbol{w} \in \mathcal{W}, \langle F(\boldsymbol{w}), \boldsymbol{w} - \boldsymbol{w}^* \rangle \geq \frac{\sigma}{\gamma}(V_{\boldsymbol{w} - \boldsymbol{w}_0}(\boldsymbol{w}^* - \boldsymbol{w}_0) + V_{\boldsymbol{w}^* - \boldsymbol{w}_0}(\boldsymbol{w} - \boldsymbol{w}_0))$.*

---

**Algorithm 1** Optimistic Dual Extrapolation

---

1: **Input:** Lipschitz constant $L > 0$ from Assumption 1, $\gamma, \delta > 0$ from Assumption 2. The VIP$(F, \mathcal{W})$ satisfying Assumption 3 ($\sigma = 0$) or Assumption 4 ($\sigma > 0$).

2: $A_0 = 0, 0 < \alpha \le \min\left\{\frac{1}{4\sqrt{2}}, \frac{\sqrt{3}}{4\sqrt{\gamma}}\right\}$.

3: $\boldsymbol{w}_0 = \boldsymbol{z}_0 \in \mathcal{W}, \boldsymbol{g}_0 = \mathbf{0}$.

4: **for** $k = 1, 2, 3, \ldots, K$ **do**

5:    $a_k = \frac{\alpha\gamma(1+\sigma A_{k-1})}{L}, A_k = A_{k-1} + a_k$.

6:    $\boldsymbol{w}_k = P_{\boldsymbol{z}_{k-1}}\left(\frac{\alpha}{L} F(\boldsymbol{w}_{k-1})\right)$.

7:    $\boldsymbol{g}_k = \boldsymbol{g}_{k-1} + a_k\left(F(\boldsymbol{w}_k) - \frac{\sigma}{\gamma}\nabla_{\boldsymbol{w}_k}\frac{1}{2}\|\boldsymbol{w}_k - \boldsymbol{w}_0\|^2\right)$.

8:    $\boldsymbol{z}_k = P_{\boldsymbol{w}_0}\left(\frac{1}{1+\sigma A_k}\boldsymbol{g}_k\right)$.

9: **end for**

10: $\tilde{\boldsymbol{w}}_K = \arg\min_{\boldsymbol{w}_k:k\in[K]}(\|\boldsymbol{w}_k - \boldsymbol{z}_{k-1}\| + \|\boldsymbol{w}_{k-1} - \boldsymbol{z}_{k-1}\|)$.

11: **return** $\tilde{\boldsymbol{w}}_K$.

---

Assumption 3 assumes the existence of weak solutions, which is also adopted in [25]. Assumption 3 is slightly weaker than the variational coherence assumption [41, 42] or the generalized monotone assumption [7]. Some nontrivial examples satisfying the generalized monotone assumption can be found in [7, 44, 28]. The generalized monotone assumption is in turn weaker than the pseudo-monotone assumption [20, 16], which is weaker than the monotone assumption (3).

**Remark 1** *In the monotone setting, the weak solution set and strong solution set are equivalent to each other; meanwhile, an approximate strong solution is also an approximate weak solution, while the reverse does not hold in general (which can explain the terms "weak" and "strong"). However, in the non-monotone setting, if the operator $F$ is continuous, a weak solution is a strong solution, while the reverse is not true in general [21, Chapter 3]. For instance, consider the minimax problem $\min_{x\in\mathbb{R}}\max_{y\in\mathbb{R}} x^2 y^2$ and let $F(x,y) = (2xy^2, -2yx^2)^T$ with $(x,y) \in \mathbb{R}^2$. Then we can verify that $(0,0)$ is the only weak solution of VIP$(F, \mathbb{R}^2)$, while the set of strong solution is the $x$-axis or the $y$-axis, and the set of Nash equilibrium is the $y$-axis.*

Assumption 4 further assumes a stronger variant of Assumption 3, which is also called as strongly variational stability in [41]. For the Euclidean setting where $\|\cdot\| := \|\cdot\|_2$ and thus $\gamma = 1$, the inequality is simplified to $\langle F(\boldsymbol{w}), \boldsymbol{w} - \boldsymbol{w}^*\rangle \ge \sigma\|\boldsymbol{w} - \boldsymbol{w}^*\|_2^2$. Assumption 4 is weaker than the strongly pseudo-monotone [19] and strongly monotone assumptions, but as we will see, is already sufficient to ensure a linear convergence rate for our method.

**Remark 2** *Our main motivation in making Assumptions 3 and 4 is to prove explicit global convergence results for VIP$(F, \mathcal{W})$ under conditions as weak as possible. However, the non-monotone subsets of Assumptions 3 and 4, a.k.a., pseudomonotone and strongly pseudomonotone respectively, also have many real applications in competitive exchange economy [2], fractional programming [10, 37], and product pricing [5]. Meanwhile, the restriction of Assumption 4 in minimization problems such as one-point convexity [23] is also used in analyzing neural networks.*

## 3 Optimistic Dual Extrapolation

In this section, we present the *optimistic dual extrapolation (OptDE)* algorithm for solving the VIP$(F, \mathcal{W})$ in (1). The method is a single-call variant of Nesterov's dual extrapolation [31]. The overall algorithm is summarized as Algorithm 1. The algorithm works under either Assumption 3 by setting $\sigma = 0$ or Assumption 4 with $\sigma > 0$.

For Algorithm 1, we define two constants $A_0$ and $\alpha$ in Step 2. Then we initialize three vectors $\boldsymbol{w}_0, \boldsymbol{z}_0$ and $\boldsymbol{g}_0$ in Step 3. In the main loop, we update the two positive numbers $a_k$ and $A_k$ in Step 5. Then we perform an "extrapolation" step in Step 6 and then "dual averaging" steps in Steps 7 and 8. As we see, as Algorithm 1 only performs one new gradient evaluation in Step 8, it is "optimistic" [36] hence the name "optimistic dual extrapolation". Once Algorithm 1 runs $K$ iterations, we return the best iterate measured by the sum of residual norms $\|\boldsymbol{w}_k - \boldsymbol{z}_{k-1}\| + \|\boldsymbol{w}_{k-1} - \boldsymbol{z}_{k-1}\|$[8].

Compared with Nesterov's dual extrapolation, the main difference is that the extrapolation Step 6 is a prox-mapping on $F(\boldsymbol{w}_{k-1})$, not on $F(\boldsymbol{z}_{k-1})$. Compared with past extra-gradient [17, 36], the main difference is that we perform dual averaging by Steps 7 and 8, instead of a "mirror descent" step. Compared with N-EG which is claimed to be a non-Euclidean extragradient method [7], not only we perform just one gradient evaluation per iteration but also do not require $\frac{1}{2}\|\cdot\|^2$ to have bounded Lipschitz continuous gradients, which is significant in the non-Euclidean setting since the norm square $\frac{1}{2}\|\cdot\|_p^2$ for $p \in (1,2)$ may not have globally bounded Lipschitz continuous gradients.

In the following, we assume $\boldsymbol{w}^*$ is a solution that satisfies Assumption 3 if $\sigma = 0$ or satisfies Assumption 4 if $\sigma > 0$.

**Theorem 1** *Let Assumptions 1 and 2 hold. For both settings $\sigma = 0$ (i.e., Assumption 3 holds) and $\sigma > 0$ (i.e., Assumption 4 holds), after $K$ iterations, Algorithm 1 returns a $\tilde{\boldsymbol{w}}_K$ such that*

$$\sup_{\boldsymbol{w}\in\mathcal{W}, \|\tilde{\boldsymbol{w}}_K - \boldsymbol{w}\| \leq D} \langle F(\tilde{\boldsymbol{w}}_K), \tilde{\boldsymbol{w}}_K - \boldsymbol{w}\rangle \leq C_0 D \|\boldsymbol{w}_0 - \boldsymbol{w}^*\| \sqrt{\frac{L}{A_{K-1} + a_1}}, \tag{10}$$

*with $C_0 = \left(1 + \frac{\delta}{\alpha\gamma}\right)\sqrt{\frac{8\alpha}{\gamma}}$, $a_1 = \frac{\alpha\gamma}{L}$, and*

$$A_{K-1} = \begin{cases} \frac{\alpha\gamma(K-1)}{L} & \text{if } \sigma = 0, \\ \frac{1}{\sigma}\left(1 + \frac{\alpha\gamma\sigma}{L}\right)^{K-1} - \frac{1}{\sigma} & \text{if } \sigma > 0. \end{cases} \tag{11}$$

*Particularly if $\sigma > 0$, we also have*

$$\|\tilde{\boldsymbol{w}}_K - \boldsymbol{w}^*\| \leq \frac{C_0}{\sigma}\|\boldsymbol{w}_0 - \boldsymbol{w}^*\|\sqrt{\frac{L}{A_{K-1} + a_1}}. \tag{12}$$

*Proof.* See Section C.4. ∎

Theorem 1 implies our main result in Table 1. As we see, for $\sigma = 0$, except for constants, our result is the same with the two-call extragradient method N-EG [7]. However, to analyze single-call methods, particularly for the setting $\sigma = 0$, the analysis is much more involved and leads to an interesting criterion of return value in Step 10 of Algorithm 1. For the setting $\sigma > 0$, then linear convergence rates can be obtained in terms of both restricted strong merit solution and solution distance. Meanwhile, for the setting $\sigma > 0$, our result in terms of restricted strong merit solution (10) can not be implied by the result of the solution distance (12), while the reverse side is true. Furthermore, when $\sigma > 0$, the result (10) is also used in deriving Corollary 1 for the monotone setting. Finally, to simplify our analysis, we did not yet optimize the constants in (10) and (12), which probably can be further improved.

In Theorem 1, we provide a unified result for the two settings $\sigma = 0$ and $\sigma > 0$ in terms of the best iterate. However, when $\sigma > 0$, we can also prove linear convergence rates in terms of last iterate, which is given in Proposition 1 below.

**Proposition 1** *Let Assumptions 1 and 2 hold. For the setting $\sigma > 0$ (i.e., Assumption 4 holds), $\forall K \geq 1$, after $K$ iterations, Algorithm 1 returns a $\boldsymbol{w}_K$ such that*

$$\sup_{\boldsymbol{w}\in\mathcal{W}, \|\boldsymbol{w}_K - \boldsymbol{w}\| \leq D} \langle F(\boldsymbol{w}_K), \boldsymbol{w}_K - \boldsymbol{w}\rangle \leq C_0 D \|\boldsymbol{w}_0 - \boldsymbol{w}^*\| \sqrt{\frac{L}{a_{K-1}}}, \tag{13}$$

*with $C_0$ defined in Theorem 1, $a_0 = a_1$ and $\forall K \geq 1$,*

$$a_K = \frac{\alpha\gamma}{L}\left(1 + \frac{\alpha\gamma\sigma}{L}\right)^{K-1}. \tag{14}$$

*Meanwhile, we also have*

$$\|\boldsymbol{w}_K - \boldsymbol{w}^*\| \leq \frac{C_0}{\sigma}\|\boldsymbol{w}_0 - \boldsymbol{w}^*\|\sqrt{\frac{L}{a_{K-1}}}. \tag{15}$$

*Proof.* See Section C.5. ∎

By Proposition 1, to prove the linear convergence of the last iterate, we do not need the strongly monotone assumption, but only Assumption 4. Despite the last iterate also has a linear convergence rate, it is slower than the rate of best iterate in Theorem 1. As we will see, Proposition 1 will also be used to prove the last iterate convergence for the monotone setting in a non-classical sense.

**Remark 3** *The motivation behind OptDE is that by generalizing Nesterov's estimation sequence, we can perform a unified convergence analysis under Assumptions 3 and 4. However, as shown in [40], if a regularizer exists, the (regularized) dual averaging steps (Steps 7 and 8 of Algorithm 1) can help us better explore the structure of regularizers such as sparsity when it exists.*

**Remark 4** *[17] has given local convergence analysis in terms of solution distance by assuming that Assumption 4 holds in a neighbourhood of the optimal solution. The analysis in [17] needs extra techniques, while the constants in the rates of [17] are implicit. Our solution distance result in (12) can be viewed as a global and explicit version of [17] by assuming Assumption 4 holds globally. Meanwhile, [17] does not give any result under Assumption 3 or in terms of restricted strong solution under Assumption 4 whereas our analysis does.*

Our results are mainly given under the coherent non-monotone Assumptions 3 and 4. As shown in Theorem 1, under Assumption 3 that includes the monotone assumption, we can obtain an $\epsilon$-accurate strong solution in $O(\epsilon^{-2})$ iterations. However, in the following we show that with *a regularization trick*, the rate can be much better in the monotone setting by using our results in Theorem 1 and Proposition 1.

First, to give our results in the monotone setting, we have Lemma 1.

**Lemma 1** *If the VIP$(F, \mathcal{W})$ is monotone, then the regularized problem VIP$(F + \epsilon \nabla \frac{1}{2\gamma}\| \cdot - \boldsymbol{w}_0\|^2, \mathcal{W})$ satisfies Assumption 4 with $\sigma = \epsilon$.*

*Proof.* See Section C.6. ∎

Due to Lemma 1, we can apply Theorem 1 and Proposition 1 to the regularized problem VIP$(F + \epsilon \nabla \frac{1}{2\gamma}\| \cdot - \boldsymbol{w}_0\|^2, \mathcal{W})$, and then obtain Corollaries 1 and 2 for the VIP$(F, \mathcal{W})$, respectively.

**Corollary 1 (Best iterate convergence in the monotone setting)** *Given $\boldsymbol{w}_0 \in \mathcal{W}$, let Assumptions 1 and 2 hold for the regularized problem VIP$(F + \epsilon \nabla \frac{1}{2\gamma}\| \cdot - \boldsymbol{w}_0\|^2, \mathcal{W})$. By optimizing the regularized problem by Algorithm 4, then the best iterate returned by Algorithm 4 satisfies*

$$
\sup_{\boldsymbol{w}\in\mathcal{W}, \|\tilde{\boldsymbol{w}}_K - \boldsymbol{w}\|\leq D, \|\boldsymbol{w}-\boldsymbol{w}_0\|\leq D} \langle F(\tilde{\boldsymbol{w}}_K), \tilde{\boldsymbol{w}}_K - \boldsymbol{w}\rangle
$$
$$
\leq \quad D\epsilon + DC_0\|\boldsymbol{w}_0 - \boldsymbol{w}^*\| \sqrt{\frac{L\epsilon}{\left(1 + \frac{\alpha\gamma\epsilon}{L}\right)^{K-1} - 1 + \frac{\alpha\gamma}{L}}},
$$

*where $C_0$ is defined in Theorem 1.*

*Proof.* See Section C.7. ∎

Compared with Theorem 1 and Proposition 1, we need an extra condition $\|\boldsymbol{w} - \boldsymbol{w}_0\| \leq D$ in Corollary 1, which can be satisfied by choosing a large enough $D$. By Corollary 1, by choosing $K = O\left(\frac{1}{\epsilon}\log\frac{1}{\epsilon}\right)$, we will obtain an $O(D\epsilon)$-accurate solution. Note that $D$ does not appear in our algorithm and is not relevant to the choice of $\epsilon$.

**Corollary 2 (Last iterate convergence in the monotone setting)** *Given $\boldsymbol{w}_0 \in \mathcal{W}$, let Assumptions 1 and 2 hold for the regularized problem VIP$(F + \epsilon \nabla \frac{1}{2\gamma}\| \cdot - \boldsymbol{w}_0\|^2, \mathcal{W})$. By optimizing the regularized problem by Algorithm 4, the last iterate of Algorithm 4 satisfies*

$$
\sup_{\boldsymbol{w}\in\mathcal{W}, \|\boldsymbol{w}_K - \boldsymbol{w}\|\leq D, \|\boldsymbol{w}-\boldsymbol{w}_0\|\leq D} \langle F(\boldsymbol{w}_K), \boldsymbol{w}_K - \boldsymbol{w}\rangle \leq D\epsilon + DC_0 L\|\boldsymbol{w}_0 - \boldsymbol{w}^*\| \sqrt{\frac{1}{\alpha\gamma\left(1 + \frac{\alpha\gamma\epsilon}{L}\right)^{K-1}}},
$$

*where $C_0$ is defined in Theorem 1.*

---

**Algorithm 2** Stochastic Optimistic Dual Extrapolation

---

1: **Input:** Lipschitz constant $L > 0$ from Assumption 1, $\gamma, \delta > 0$ from Assumption 2. The VIP$(F, \mathcal{W})$ satisfying Assumption 3 ($\sigma = 0$) or Assumption 4 ($\sigma > 0$).
2: $A_0 = 0, \alpha = \min\{\frac{\gamma}{32}, \frac{1}{16}\}$.
3: $\boldsymbol{w}_0 = \boldsymbol{z}_0 \in \mathcal{W}, \boldsymbol{g}_0 = \boldsymbol{0}$.
4: **for** $k = 1, 2, 3, \ldots, K$ **do**
5:     $a_k = \frac{\alpha\gamma\sqrt{1+\sigma A_{k-1}}}{L}, A_k = A_{k-1} + a_k$.
6:     $\boldsymbol{w}_k = P_{\boldsymbol{z}_{k-1}}\big(\frac{\alpha^2\gamma}{L^2 a_k} F(\boldsymbol{w}_{k-1}; \xi_{k-1})\big)$.
7:     $\boldsymbol{g}_k = \boldsymbol{g}_{k-1} + a_k\big(F(\boldsymbol{w}_k; \xi_k) - \frac{\sigma}{\gamma}\nabla_{\boldsymbol{w}_k}\frac{1}{2}\|\boldsymbol{w}_k - \boldsymbol{w}_0\|^2\big)$.
8:     $\boldsymbol{z}_k = P_{\boldsymbol{w}_0}\big(\frac{1}{1+\sigma A_k}\boldsymbol{g}_k\big)$.
9: **end for**
10: $\tilde{\boldsymbol{w}}_K = \boldsymbol{w}_k$, where $k$ is chosen at random with probability distribution $\{\frac{a_1}{A_K}, \frac{a_2}{A_K}, \ldots, \frac{a_K}{A_K}\}$.
11: **return** $\tilde{\boldsymbol{w}}_K$.

---

*Proof.* See Section C.8. ∎

Similar to Corollary 1 for best iterate, in Corollary 2, by choosing $K = O\big(\frac{1}{\epsilon}\log\frac{1}{\epsilon}\big)$, the last iterate will be an $O(D\epsilon)$-accurate strong solution, which is significantly better than the tight bound $O(1/\epsilon^2)$ for last iterate [13]. Nevertheless, it should be noted that Corollary 2 is in a non-classical sense: we do not guarantee last iterate convergence for all $K \geq 1$, but only after $K = O\big(\frac{1}{\epsilon}\log\frac{1}{\epsilon}\big)$ with a prescribed accuracy parameter $\epsilon$. Thus our result does not contradict with the lower bound of last iterate [13].

Meanwhile, our proof only relies on the regularized problem VIP$(F + \epsilon\nabla\frac{1}{2\gamma}\|\cdot - \boldsymbol{w}_0\|^2, \mathcal{W})$ satisfying Assumption 4 with $\sigma = \epsilon$, which holds if the VIP$(F, \mathcal{W})$ is monotone. However, it is not necessary for the VIP$(F, \mathcal{W})$ to be monotone. For instance, if the VIP$(F, \mathcal{W})$ satisfies Assumption 3 and $\boldsymbol{w}_0 = \boldsymbol{w}^*$, then the VIP$(F + \epsilon\nabla\frac{1}{2\gamma}\|\cdot - \boldsymbol{w}_0\|^2, \mathcal{W})$ also satisfies Assumption 4 with $\sigma = \epsilon$. Of course, letting $\boldsymbol{w}_0 = \boldsymbol{w}^*$ is impractical and we leave the more general setting of $\boldsymbol{w}_0$ under non-monotone settings for further research.

**Remark 5** *Recently, [9] has proposed a different Halpern iteration method under the monotone and Lipschitz assumptions. The Halpern iteration method does not need to know the Lipschitz constant and thus is parameter-free, and also attains the $O\big(\frac{1}{\epsilon}\log\frac{1}{\epsilon}\big)$ convergence rate. Nevertheless, there are two major differences: The Halpern iteration method has two-loop, while our OptDE method is a single-loop single-call method; now the Halpern iteration method is limited to the Euclidean setting, while ours can have theoretical guarantees in the non-Euclidean setting.*

## 4 Stochastic Optimistic Dual Extrapolation

In this section, we present a stochastic version of the above OptDE method, *a.k.a., stochastic optimistic dual extrapolation (SOptDE)*, which is given in Algorithm 2. Compared with the OptDE method in Algorithm 1, the main difference is that Algorithm 2 approximates $\{F(\boldsymbol{w}_k)\}$ by the unbiased stochastic estimations $\{F(\boldsymbol{w}_k; \xi_k)\}$, where the randomness is from the *i.i.d* random variables $\{\xi_k\}$. For simplicity, in this section, we use $\mathbb{E}_\xi[\cdot]$ to denote the expectation *w.r.t.* $\xi$ while fixing the previous randomness; meanwhile, we use $\mathbb{E}[\cdot]$ to denote the expectation *w.r.t.* the randomness of all the history. Formally, we make Assumption 5.

**Assumption 5** $\forall \boldsymbol{w} \in \mathcal{W}$, $F(\boldsymbol{w}; \xi)$ *is an unbiased estimation of* $F(\boldsymbol{w})$ *such that* $\mathbb{E}_\xi[F(\boldsymbol{w}; \xi)] = F(\boldsymbol{w})$; *meanwhile the variance of* $F(\boldsymbol{w}; \xi)$ *is bounded by* $s^2$ *such that* $\mathbb{E}_\xi[\|F(\boldsymbol{w}; \xi) - F(\boldsymbol{w})\|_*^2] \leq s^2$.

Meanwhile, to cancel the error from randomness, in Algorithm 2, when $\sigma > 0$, we consider a more conservative parameter setting $a_k = \frac{\alpha\gamma\sqrt{1+\sigma A_{k-1}}}{L}$ rather than $a_k = \frac{\alpha\gamma(1+\sigma A_{k-1})}{L}$ of Algorithm 1. Furthermore, because of the randomness, choosing the exact best iterate as in the deterministic case is no longer meaningful as its expectation is impossible to compute. In this case, we choose $\tilde{\boldsymbol{w}}_K$ at random according to the distribution $\{\frac{a_1}{A_K}, \frac{a_2}{A_K}, \ldots, \frac{a_K}{A_K}\}$, which also facilitates theoretical analysis[9].

**Theorem 2** *For the setting $\sigma = 0$ (i.e., Assumption 3 holds), after $K$ iterations, Algorithm 5 returns a $\tilde{\boldsymbol{w}}_K$ such that*

$$\mathbb{E}\Big[\sup_{\boldsymbol{w}\in\mathcal{W}, \|\tilde{\boldsymbol{w}}_K - \boldsymbol{w}\|\leq D}\langle F(\tilde{\boldsymbol{w}}_K), \tilde{\boldsymbol{w}}_K - \boldsymbol{w}\rangle\Big]$$

$$\leq \sqrt{2}(1+\delta)LD\sqrt{\frac{\|\boldsymbol{w}^* - \boldsymbol{w}_0\|^2}{8\alpha\gamma K} + \frac{s^2}{L^2}} + L^2\Big(\frac{\|\boldsymbol{w}^* - \boldsymbol{w}_0\|^2}{8\alpha\gamma K} + \frac{s^2}{L^2}\Big) + \frac{s^2}{2L^2}. \qquad (16)$$

*For $\sigma > 0$, (i.e., Assumption 4 holds), we have*

$$\mathbb{E}[\|\tilde{\boldsymbol{w}}_K - \boldsymbol{w}^*\|^2] \leq \frac{32L^2}{\sigma^2(\alpha\gamma)^2(K+1)^2}\Big(\frac{8\alpha s^2 K}{L^2} + \frac{1}{2\gamma}\|\boldsymbol{w}^* - \boldsymbol{w}_0\|^2\Big). \qquad (17)$$

*Proof.* See Section D.4. ∎

As show in (16), for the setting $\sigma = 0$ (*a.k.a.*, Assumption 3) even if the number of iterations $K \to \infty$, the expected restricted strong merit function can only be upper bounded by $O\big(\frac{s}{L}\big)$. Thus to guarantee the convergence of SOptDE, the variance should be $o(1)$, such as $s^2 = O\big(\frac{1}{K}\big)$. In the Euclidean setting that $\|\cdot\| := \|\cdot\|_2$, by the concentration inequality [39], to attain a variance of $O\big(\frac{1}{K}\big)$, we need $O(K)$ samples. Thus combining the setting $s^2 = O\big(\frac{1}{K}\big)$ and the result in (16), it can be verified that the single-call SOptDE method needs $O(1/\epsilon^4)$ number of samples to obtain an $\epsilon$-accurate solution in terms of the expected restricted strong merit function.

To develop the two-call stochastic extragradient method SEG [18] under the pseudomonotone assumption[10], [18] has also considered variance reduction with a large batch size and used a "quadratic natural residual" (in our notation, it is $\mathbb{E}[\|\boldsymbol{w}_k - \boldsymbol{z}_{k-1}\|^2]$) to measure the accuracy, which in turns can be used to derive the same complexity result $O(1/\epsilon^4)$ as SOptDE in terms of expected restricted strong merit function (see the supplementary material). OSG [26] is a single-call version of SEG, which also uses quadratic natural residual as a convergence measure. However in the general constrained setting, it is not know how to convert the guarantee of quadratic natural residual of the single-call OSG into the guarantee of expected restricted strong merit function. In fact, in our single-call setting, the "(quadratic) natural residual" $\mathbb{E}[\|\boldsymbol{w}_k - \boldsymbol{z}_{k-1}\|^2]$ is no longer useful in deriving the theoretical guarantee by expected restricted strong merit function. As a result, we consider the term $\mathbb{E}[\|\boldsymbol{w}_k - \boldsymbol{z}_{k-1}\|^2 + \|\boldsymbol{w}_{k-1} - \boldsymbol{z}_{k-1}\|^2]$, which makes our proof quite different from that in [18].

Under the stronger Assumption 4, our result is given in terms of the expected solution distance. As shown in (17), under Assumption 4, SOptDE can converge provably even when the variance $s^2$ is constant. In fact, the $O\big(\frac{1}{K}\big)$ is optimal and has been obtained by the two-call extragradient method ESA [19] under the pseudomonotone assumption. Meanwhile, [43] used the plain stochastic gradient descent algorithm and obtained the $O\big(\frac{1}{K}\big)$ result for strongly monotone variational inequalities, which can also be extended to the setting that $\sigma$-weak solution exists.

With the aggressive parameter setting $a_k = \frac{\alpha\gamma(1+\sigma A_{k-1})}{L}$ and a large batch size strategy, we also obtain the first convergence guarantee $O(1/\epsilon^2 \log\frac{1}{\epsilon})$ in terms of restricted strong merit function as shown in Table 2 (see details in the supplementary material).

## 5 Concluding Remarks

In this paper, we proposed a single-call extragradient method *optimistic dual extrapolation (OptDE)* beyond the monotone setting and also extended it to the stochastic setting as *stochastic optimistic dual extrapolation (SOptDE)*. We systematically proved the convergence results of OptDE and SOptDE under the Assumption 3 that a weak solution exists and Assumption 4 that a strongly weak solution exists. We also show beneficial implications of our analysis in both non-monotone and monotone settings. In the future, we will further study how the proposed new methods may lead to improved computational efficiency and performance guarantees in a wide range of machine learning problems such as the training of adversarial deep neural networks.

## Broader Impact

In this paper, we discuss a systematic theoretical analysis for single-call extragradient methods, which has been widely used for modern machine learning applications. The theoretical results in this paper can bring in meaningful insight and understanding for practical algorithms.

## Acknowledgement

Chaobing and Yi acknowledge support from Tsinghua-Berkeley Shenzhen Institute (TBSI) Research Fund. Yichao and Yi acknowledge funding from Sony Research. Yi acknowledges support from ONR grant N00014-20-1-2002 and the joint Simons Foundation-NSF DMS grant #2031899, as well as support from Berkeley AI Research (BAIR), Berkeley FHL Vive Center for Enhanced Reality, and Berkeley Center for Augmented Cognition.

## Footnotes

[2]Recently, [13] shows the first tight last iterate result for general smooth convex-concave minimax problems with Lipschitz derivatives of operators.

[3]As we will see, it is a localized version of our assumption.

[4]For given a number of iterations, the best iterate can be explicitly found and happen before the last iterate.

[5]Here the last iterate is not in the classical sense, which will be explained in Section 3.

[6]The results of the SEG [18], ESA [19] algorithms are given under pseudomonotone and strongly pseudomonotone assumptions respectively, which are slightly stronger than our assumptions.

[7]The original result of SEG is given by "square natural residual", which can be used to derive the strong solution guarantee in Table 2 (see the supplementary material for detail).

[8]This return value is given according to our convergence analysis.

[9]In practice, nevertheless, one may often consider choosing the last iterate for simplicity.

[10]We can verify that the result in [18] can be extended under our Assumption 3.

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
