[Supplementary Material]

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

# A Convergence Analysis of Optimistic Dual Extrapolation

Based on the optimality condition of $z_k$ and Assumption 2, we have Lemma 2.

**Lemma 2** *In Algorithm 1, let*

$$E_{1k} := a_k \Big\langle F(\boldsymbol{w}_k) + \frac{L}{\alpha\gamma}\nabla_{\boldsymbol{w}_k}\frac{1}{2}\|\boldsymbol{w}_k - \boldsymbol{z}_{k-1}\|^2, \boldsymbol{w}_k - \boldsymbol{z}_k \Big\rangle$$
$$- \frac{L}{\alpha\gamma}\Big(\frac{1}{2}\|\boldsymbol{w}_k - \boldsymbol{z}_{k-1}\|^2 + \frac{\gamma}{2}\|\boldsymbol{w}_k - \boldsymbol{z}_k\|^2\Big), \tag{18}$$

*then $\forall \boldsymbol{u}, \boldsymbol{w}_0 \in \mathcal{W}$, we have*

$$\sum_{k=1}^{K} a_k\left(\langle F(\boldsymbol{w}_k), \boldsymbol{w}_k - \boldsymbol{u}\rangle - \frac{\sigma}{\gamma}V_{\boldsymbol{w}_k-\boldsymbol{w}_0}(\boldsymbol{u} - \boldsymbol{w}_0)\right) \leq \sum_{k=1}^{K}E_{1k} + \frac{1}{2\gamma}\|\boldsymbol{u} - \boldsymbol{w}_0\|^2. \tag{19}$$

*Proof.* See Section C.1. ∎

In Lemma 2, the sequence $\{E_{1k}\}$ can be viewed as the errors we need to bound in each iteration. The upper bound of the sum of $\{E_{1k}\}$ is given in Lemma 3 below.

**Lemma 3** *In Algorithm 1, $\forall k \in [K]$, we have*

$$\sum_{k=1}^{K}E_{1k} \leq -\frac{L}{8\alpha}\sum_{k=1}^{K}a_{k-1}(\|\boldsymbol{w}_k - \boldsymbol{z}_{k-1}\|^2 + \|\boldsymbol{w}_{k-1} - \boldsymbol{z}_{k-1}\|^2),$$

*where we define $a_0 := a_1$ for convenience.*

*Proof.* See Section C.2. ∎

By Lemma 3, $\forall\, 0 < \alpha \leq \min\left\{\frac{1}{4\sqrt{2}}, \frac{\sqrt{3}}{4\sqrt{\gamma}}\right\}$ and $k \in [K]$, $\sum_{k=1}^{K}E_{1k}$ is upper bounded by the sum of strictly negative terms about $\|\boldsymbol{w}_k - \boldsymbol{z}_{k-1}\|^2 + \|\boldsymbol{w}_{k-1} - \boldsymbol{z}_{k-1}\|^2$, which makes it possible to give a upper bound about $\min_{k\in[K]}(\|\boldsymbol{w}_k - \boldsymbol{z}_{k-1}\| + \|\boldsymbol{w}_{k-1} - \boldsymbol{z}_{k-1}\|)$. To show the guarantees by restricted strong merit function and the distance $\|\boldsymbol{w}_k - \boldsymbol{w}^*\|$, we give Lemma 4.

**Lemma 4** *In Algorithm 1, $\forall k \in [K]$, we have,*

$$\sup_{\boldsymbol{w}\in\mathcal{W}, \|\boldsymbol{w}_k-\boldsymbol{w}\|\leq D}\langle F(\boldsymbol{w}_k), \boldsymbol{w}_k - \boldsymbol{w}\rangle \leq \big(1 + \frac{\delta}{\alpha\gamma}\big)DL(\|\boldsymbol{w}_k - \boldsymbol{z}_{k-1}\| + \|\boldsymbol{z}_{k-1} - \boldsymbol{w}_{k-1}\|). \tag{20}$$

*If $\sigma > 0$, then we have*

$$\|\boldsymbol{w}_k - \boldsymbol{w}^*\| \leq \big(1 + \frac{\delta}{\alpha\gamma}\big)\frac{L}{\sigma}(\|\boldsymbol{w}_k - \boldsymbol{z}_{k-1}\| + \|\boldsymbol{z}_{k-1} - \boldsymbol{w}_{k-1}\|). \tag{21}$$

*Proof.* See Section C.3. ∎

Then combining Lemmas 2, 3 and 4, we obtain Theorem 1 in main body (see Section C.4 for the proof.).

# B Convergence Analysis of Stochastic Optimistic Dual Extrapolation

We can extend the proof for the OptDE method in Section A to the stochastic setting for Lemmas 5, 6 and 7 and then obtain Theorems 2. First, we extend Lemma 2 into Lemma 5.

**Lemma 5** *In Algorithm 5, $\forall k \in [K]$, we have the following inequality: $\forall \boldsymbol{u}, \boldsymbol{w}_0 \in \mathcal{W}$, let*

$$E_{2k} := a_k\Big\langle F(\boldsymbol{w}_k; \xi_k) + \frac{L^2 a_k}{(\alpha\gamma)^2}\nabla_{\boldsymbol{w}_k}\frac{1}{2}\|\boldsymbol{w}_k - \boldsymbol{z}_{k-1}\|^2, \boldsymbol{w}_k - \boldsymbol{z}_k \Big\rangle$$
$$- \frac{L^2 a_k}{(\alpha\gamma)^2}\Big(\frac{1}{2}\|\boldsymbol{w}_k - \boldsymbol{z}_{k-1}\|^2 + \frac{\gamma}{2}\|\boldsymbol{w}_k - \boldsymbol{z}_k\|^2\Big) + a_k\langle F(\boldsymbol{w}_k) - F(\boldsymbol{w}_k; \xi_k), \boldsymbol{w}_k - \boldsymbol{u}\rangle, \tag{22}$$

*then we have*

$$\mathbb{E}\left[\sum_{k=1}^{K} a_k\left(\langle F(\boldsymbol{w}_k), \boldsymbol{w}_k - \boldsymbol{u}\rangle - \frac{\sigma}{\gamma}V_{\boldsymbol{w}_k-\boldsymbol{w}_0}(\boldsymbol{u}-\boldsymbol{w}_0)\right)\right] \leq \mathbb{E}\left[\sum_{k=1}^{K} E_{2k}\right] + \frac{1}{2\gamma}\|\boldsymbol{u}-\boldsymbol{w}_0\|^2. \quad (23)$$

*Proof.* See Section D.1. ∎

Compared with the $E_{1k}$ of Lemma 2, $E_{2k}$ contains an extra term $a_k\langle F(\boldsymbol{w}_k) - F(\boldsymbol{w}_k;\xi_k), \boldsymbol{w}_k - \boldsymbol{u}\rangle$. Then based on the definition of $E_{2k}$ and Assumption 5, we have Lemma 6.

**Lemma 6** *In Algorithm 5, $\forall k \in [K]$ and $\forall \boldsymbol{u} \in \mathcal{W}$, we have*

$$\mathbb{E}\left[\sum_{k=1}^{K} E_{2k}\right] \leq -\mathbb{E}\left[4\alpha\sum_{k=1}^{K}(\|\boldsymbol{w}_k-\boldsymbol{z}_{k-1}\|^2 + \|\boldsymbol{w}_{k-1}-\boldsymbol{z}_{k-1}\|^2)\right] + \frac{8\alpha s^2 K}{L^2}. \quad (24)$$

*Proof.* See Section D.2. ∎

Lemma 6 extends Lemma 3 into the stochastic setting. Meanwhile, by the optimality condition of $\boldsymbol{w}_k$, and Assumptions 1, 2 and 5, we can extend Lemma 4 to Lemma 7.

**Lemma 7** *In Algorithm 5, for the setting $\sigma = 0$ and $\forall k \in [K]$, we have,*

$$\mathbb{E}_{\xi_{k-1}}\left[\sup_{\boldsymbol{w}\in\mathcal{W},\|\boldsymbol{w}_k-\boldsymbol{w}\|\leq D}\langle F(\boldsymbol{w}_k), \boldsymbol{w}_k - \boldsymbol{w}\rangle\right]$$

$$\leq \left(1 + \frac{\delta}{\alpha\gamma}\right)LD\mathbb{E}_{\xi_{k-1}}[(\|\boldsymbol{w}_k-\boldsymbol{z}_{k-1}\| + \|\boldsymbol{w}_{k-1}-\boldsymbol{z}_{k-1}\|)] + \frac{L^2}{2}\mathbb{E}_{\xi_{k-1}}[\|\boldsymbol{w}_k-\boldsymbol{w}_{k-1}\|^2] + \frac{s^2}{2L^2}.$$

*Proof.* See Section D.3. ∎

Then combining Lemmas 5, 6 and 7, we obtain Theorem 2 for the SOptDE method in the main body (see Section D.4 for the proof).

## B.1 The $O\left(\frac{1}{\epsilon^2}\log\frac{1}{\epsilon}\right)$ rate in terms of restricted strong merit function under Assumption 4

It turns out that with the conservative setting $a_k = \frac{\alpha\gamma\sqrt{1+\sigma A_{k-1}}}{L}$, we can not obtain strong convergence results in terms of restricted strong merit function. To obtain the rate $O\left(\frac{1}{\epsilon^2}\log\frac{1}{\epsilon}\right)$, we need adopt the more aggressive setting $a_k = \frac{\alpha\gamma(1+\sigma A_{k-1})}{L}$ with a large batch size strategy, which is given in Algorithm 3. With this setting, we have Proposition 2.

---

**Algorithm 3** Stochastic Optimistic Dual Extrapolation (**Version 2**)

---

1: **Input:** Lipshitz constant $L > 0$ from Assumption 1, $\gamma, \delta > 0$ from Assumption 2. The VIP$(F, \mathcal{W})$ satisfying Assumption 3 ($\sigma = 0$) or Assumption 4 ($\sigma > 0$).
2: $A_0 = 0, 0 < \alpha \leq \min\left\{\frac{1}{8}, \frac{\sqrt{3}}{4\sqrt{2}\gamma}\right\}$.
3: $\boldsymbol{w}_0 = \boldsymbol{z}_0 \in \mathcal{W}, \boldsymbol{g}_0 = \boldsymbol{0}$.
4: **for** $k = 1, 2, 3, \ldots, K$ **do**
5:    $a_k = \frac{\alpha\gamma(1+\sigma A_{k-1})}{L}, A_k = A_{k-1} + a_k$.
6:    $\boldsymbol{w}_k = \arg\min_{\boldsymbol{w}\in\mathcal{W}}\left\{\langle F(\boldsymbol{w}_{k-1};\xi_{k-1}), \boldsymbol{w}\rangle + \frac{L}{2\alpha\gamma}\|\boldsymbol{w}-\boldsymbol{z}_{k-1}\|^2\right\}$.
7:    $\boldsymbol{g}_k = \boldsymbol{g}_{k-1} + a_k\left(F(\boldsymbol{w}_k;\xi_k) - \frac{\sigma}{\gamma}\nabla_{\boldsymbol{w}_k}\frac{1}{2}\|\boldsymbol{w}_k-\boldsymbol{w}_0\|^2\right)$.
8:    $\boldsymbol{z}_k = \arg\min_{\boldsymbol{z}\in\mathcal{W}}\left\{\langle \boldsymbol{g}_k, \boldsymbol{z}\rangle + \frac{1+\sigma A_k}{2\gamma}\|\boldsymbol{z}-\boldsymbol{w}_0\|^2\right\}$
9: **end for**
10: $\tilde{\boldsymbol{w}}_K = \boldsymbol{w}_k$, where $k$ is chosen at random with probability distribution $\left\{\frac{a_1}{A_K}, \frac{a_2}{A_K}, \ldots, \frac{a_K}{A_K}\right\}$.
11: **return** $\tilde{\boldsymbol{w}}_K$.

---

**Proposition 2** *Let Assumptions 1, 2, 4 and 5 hold. After $K$ iterations, Algorithm 3 returns a $\tilde{\boldsymbol{w}}_K$ such that*

$$\mathbb{E}\Big[\sup_{\boldsymbol{w}\in\mathcal{W},\|\tilde{\boldsymbol{w}}_K-\boldsymbol{w}\|\leq D}\langle F(\tilde{\boldsymbol{w}}_K),\tilde{\boldsymbol{w}}_K-\boldsymbol{w}\rangle\Big]$$

$$\leq C_1 LD\sqrt{\frac{\|\boldsymbol{w}^*-\boldsymbol{w}_0\|^2}{L(A_{K-1}+a_1)}+\frac{s^2}{L^2}}+L^2\Big(\frac{\|\boldsymbol{w}^*-\boldsymbol{w}_0\|^2}{L(A_{K-1}+a_1)}+\frac{s^2}{L^2}\Big)+\frac{s^2}{2L^2}. \tag{25}$$

*with $C_1 := 4\big(1+\frac{\delta}{\alpha\gamma}\big)\sqrt{\frac{\alpha}{\gamma}}$, $a_1 = \frac{\alpha\gamma}{L}$, and*

$$A_{K-1}=\frac{1}{\sigma}\Big(1+\frac{\alpha\gamma\sigma}{L}\Big)^{K-1}-\frac{1}{\sigma}. \tag{26}$$

The proof of Proposition 2 follows the same pipeline of proving Theorem 2, except that we use the setting $a_k = \frac{\alpha\gamma(1+\sigma A_{k-1})}{L}$ that is also used in Algorithm 4. We leave the proof of Proposition 2 as a simple exercise.

In Proposition 2, if we hope the variance of the stochastic estimation $\{F(\boldsymbol{w}_k;\xi_k)\}$ as $s^2 = O(\frac{1}{A_{K-1}+a_1})$, then we need $O(A_{K-1}+a_1)$ stochastic samples per iteration. Meanwhile, to attain an expected $\epsilon$-accurate strong solution, we will need $O(\log\frac{1}{\epsilon})$ number of iterations. Thus the total number of stochastic samples we need is $O(\frac{1}{\epsilon^2}\log\frac{1}{\epsilon})$.

## B.2 The " (quadratic) natural residual function" [18] and restricted strong merit function

In our notation, for any $\boldsymbol{w}\in\mathcal{W}$, the (quadratic) natural residual function in [18] is defined by: given $\eta>0$,

$$r_\eta(\boldsymbol{w})=\|\boldsymbol{w}-P_{\boldsymbol{w}}(\eta F(\boldsymbol{w}))\|^2, \tag{27}$$

which can be used to derive the restricted strong merit function as Proposition 3.

**Proposition 3** *Let $\boldsymbol{w}' := P_{\boldsymbol{w}}(\eta F(\boldsymbol{w})) = \arg\min_{\boldsymbol{z}\in\mathcal{W}}\{\langle \eta F(\boldsymbol{w}),\boldsymbol{z}\rangle+\frac{1}{2\gamma}\|\boldsymbol{z}-\boldsymbol{w}\|^2\}$. Then we have*

$$\sup_{\boldsymbol{z}\in\mathcal{W},\|\boldsymbol{w}'-\boldsymbol{z}\|\leq D}\langle F(\boldsymbol{w}'),\boldsymbol{w}'-\boldsymbol{z}\rangle\leq\Big(L+\frac{\delta}{\eta\gamma}\Big)D\sqrt{r_\eta(\boldsymbol{w})}. \tag{28}$$

*Proof.* It follows that

$$
\begin{aligned}
\langle F(\boldsymbol{w}'),\boldsymbol{w}'-\boldsymbol{z}\rangle &= \langle F(\boldsymbol{w}')-\big(F(\boldsymbol{w})+\frac{1}{\eta}\nabla_{\boldsymbol{w}'}\frac{1}{2\gamma}\|\boldsymbol{w}'-\boldsymbol{w}\|^2\big),\boldsymbol{w}'-\boldsymbol{z}\rangle\\
&\quad +\Big\langle F(\boldsymbol{w})+\frac{1}{\eta}\nabla_{\boldsymbol{w}'}\frac{1}{2\gamma}\|\boldsymbol{w}'-\boldsymbol{w}\|^2,\boldsymbol{w}'-\boldsymbol{z}\Big\rangle\\
&\overset{(a)}{\leq} \langle F(\boldsymbol{w}')-\big(F(\boldsymbol{w})+\frac{1}{\eta}\nabla_{\boldsymbol{w}'}\frac{1}{2\gamma}\|\boldsymbol{w}'-\boldsymbol{w}\|^2\big),\boldsymbol{w}'-\boldsymbol{z}\rangle\\
&\overset{(b)}{\leq} \|F(\boldsymbol{w}')-F(\boldsymbol{w})\|_*\|\boldsymbol{w}'-\boldsymbol{z}\|+\frac{1}{\eta}\big\|\nabla_{\boldsymbol{w}'}\frac{1}{2\gamma}\|\boldsymbol{w}'-\boldsymbol{w}\|^2\big\|_*\|\boldsymbol{w}'-\boldsymbol{z}\|\\
&\overset{(c)}{\leq} L\|\boldsymbol{w}'-\boldsymbol{w}\|\|\boldsymbol{w}'-\boldsymbol{z}\|+\frac{1}{\eta\gamma}\delta\|\boldsymbol{w}'-\boldsymbol{w}\|\|\boldsymbol{w}'-\boldsymbol{z}\|\\
&= \Big(L+\frac{\delta}{\eta\gamma}\Big)\|\boldsymbol{w}'-\boldsymbol{w}\|\|\boldsymbol{w}'-\boldsymbol{z}\|.
\end{aligned} \tag{29}
$$

where $(a)$ is by the optimality condition of $\boldsymbol{w}'$, $(b)$ is by the Cauchy-Schwarz inequality, $(c)$ is by the Lipschitz continuity of $F(\mathbf{w})$ and the bounded assumption (7). So we have

$$\sup_{\boldsymbol{z}\in\mathcal{W},\|\boldsymbol{w}'-\boldsymbol{z}\|\leq D}\langle F(\boldsymbol{w}'),\boldsymbol{w}'-\boldsymbol{z}\rangle\leq\Big(L+\frac{\delta}{\eta\gamma}\Big)D\|\boldsymbol{w}'-\boldsymbol{w}\|=\Big(L+\frac{\delta}{\eta\gamma}\Big)D\sqrt{r_\eta(\boldsymbol{w})}. \tag{30}$$

∎

## C  Proof of Section A

By the definition of proximal operator (8), we can equivalently reformulate the optimistic dual extrapolation (OptDE) algorithm in the main body as Algorithm 4. Then based on the definition of $\boldsymbol{g}_k$ in Step 7 and the definition of the Bregman divergence $V_{\boldsymbol{w}}(\boldsymbol{u})(\boldsymbol{w}, \boldsymbol{u} \in \mathcal{W})$, we can verify that

$$\boldsymbol{z}_k = \arg\min_{\boldsymbol{z} \in \mathcal{W}} \left\{ \psi_k(\boldsymbol{z}) := \sum_{i=1}^{k} a_i \Big( \langle F(\boldsymbol{w}_i), \boldsymbol{z} - \boldsymbol{u} \rangle + \frac{\sigma}{\gamma} V_{\boldsymbol{w}_i - \boldsymbol{w}_0}(\boldsymbol{z} - \boldsymbol{w}_0) \Big) + \frac{1}{2\gamma} \|\boldsymbol{z} - \boldsymbol{w}_0\|^2 \right\}, \quad (31)$$

where $\boldsymbol{u}$ is an arbitrary vector in $\mathcal{W}$ and is irrelevant to the minimizer $\boldsymbol{z}_k$. In our context, $\psi_k(\boldsymbol{z})$ plays the role of a "generalized estimation sequence" to help us conduct convergence analysis. By the $\gamma$-strong convexity of the Bregman divergence $V_{\boldsymbol{w}_i - \boldsymbol{w}_0}(\boldsymbol{z} - \boldsymbol{w}_0)$, we know that $\psi_k(\boldsymbol{z})$ is strongly convex with strong convexity parameter $1 + \sigma \sum_{i=1}^{k} a_i = 1 + \sigma A_k$.

---

**Algorithm 4** Optimistic Dual Extrapolation (**Reformulation**)

---

1: **Input:** Lipschitz constant $L > 0$ from Assumption 1, $\gamma, \delta > 0$ from Assumption 2. The VIP$(F, \mathcal{W})$ satisfying Assumption 3 ($\sigma = 0$) or Assumption 4 ($\sigma > 0$).
2: $A_0 = 0, 0 < \alpha \leq \min\left\{ \frac{1}{4\sqrt{2}}, \frac{\sqrt{3}}{4\sqrt{\gamma}} \right\}$.
3: $\boldsymbol{w}_0 = \boldsymbol{z}_0 \in \mathcal{W}$.
4: **for** $k = 1, 2, 3, \ldots, K$ **do**
5:  $a_k = \frac{\alpha\gamma(1 + \sigma A_{k-1})}{L}, A_k = A_{k-1} + a_k$.
6:  $\boldsymbol{w}_k = \arg\min_{\boldsymbol{w} \in \mathcal{W}} \left\{ \langle F(\boldsymbol{w}_{k-1}), \boldsymbol{w} \rangle + \frac{L}{2\alpha\gamma} \|\boldsymbol{w} - \boldsymbol{z}_{k-1}\|^2 \right\}$.
7:  $\boldsymbol{g}_k = \boldsymbol{g}_{k-1} + a_k \left( F(\boldsymbol{w}_k) - \frac{\sigma}{\gamma} \nabla_{\boldsymbol{w}_k} \frac{1}{2} \|\boldsymbol{w}_k - \boldsymbol{w}_0\|^2 \right)$.
8:  $\boldsymbol{z}_k = \arg\min_{\boldsymbol{z} \in \mathcal{W}} \left\{ \langle \boldsymbol{g}_k, \boldsymbol{z} \rangle + \frac{1 + \sigma A_k}{2\gamma} \|\boldsymbol{z} - \boldsymbol{w}_0\|^2 \right\}$.
9: **end for**
10: $\tilde{\boldsymbol{w}}_K = \arg\min_{\boldsymbol{w}_k : k \in [K]} (\|\boldsymbol{w}_k - \boldsymbol{z}_{k-1}\| + \|\boldsymbol{w}_{k-1} - \boldsymbol{z}_{k-1}\|)$.
11: **return** $\tilde{\boldsymbol{w}}_K$.

---

### C.1  Proof of Lemma 2

*Proof.* Given the definition of the generalized estimation sequence $\psi_k(\boldsymbol{z})$ in (31) and the minimizer $\boldsymbol{z}_k$ in Algorithm 4, by the optimality condition of $\boldsymbol{z}_k$, we have: $\forall \boldsymbol{u} \in \mathcal{W}$,

$$\left\langle \sum_{i=1}^{k} a_i \big( F(\boldsymbol{w}_i) + \frac{\sigma}{\gamma} \nabla V_{\boldsymbol{w}_i - \boldsymbol{w}_0}(\boldsymbol{z}_k - \boldsymbol{w}_0) \big) + \nabla_{\boldsymbol{z}_k} \frac{1}{2\gamma} \|\boldsymbol{z}_k - \boldsymbol{w}_0\|^2, \boldsymbol{u} - \boldsymbol{z}_k \right\rangle \geq 0. \quad (32)$$

Then we have $\forall k \in [K]$,

$$\begin{aligned}
\psi_k(\boldsymbol{z}_k) &= \sum_{i=1}^{k} a_i \left( \langle F(\boldsymbol{w}_i), \boldsymbol{z}_k - \boldsymbol{u} \rangle + \frac{\sigma}{\gamma} V_{\boldsymbol{w}_i - \boldsymbol{w}_0}(\boldsymbol{z}_k - \boldsymbol{w}_0) \right) + \frac{1}{2\gamma} \|\boldsymbol{z}_k - \boldsymbol{w}_0\|^2 \\
&\overset{(a)}{\leq} \frac{\sigma}{\gamma} \sum_{i=1}^{k} a_i \left( \langle \nabla V_{\boldsymbol{w}_i - \boldsymbol{w}_0}(\boldsymbol{z}_k - \boldsymbol{w}_0), \boldsymbol{u} - \boldsymbol{z}_k \rangle + V_{\boldsymbol{w}_i - \boldsymbol{w}_0}(\boldsymbol{z}_k - \boldsymbol{w}_0) \right) \\
&\quad + \left\langle \nabla_{\boldsymbol{z}_k} \frac{1}{2\gamma} \|\boldsymbol{z}_k - \boldsymbol{w}_0\|^2, \boldsymbol{u} - \boldsymbol{z}_k \right\rangle + \frac{1}{2\gamma} \|\boldsymbol{z}_k - \boldsymbol{w}_0\|^2 \\
&\overset{(b)}{\leq} \frac{\sigma}{\gamma} \sum_{i=1}^{k} a_i V_{\boldsymbol{w}_i - \boldsymbol{w}_0}(\boldsymbol{u} - \boldsymbol{w}_0) + \frac{1}{2\gamma} \|\boldsymbol{u} - \boldsymbol{w}_0\|^2, \quad (33)
\end{aligned}$$

where $(a)$ is by the optimality condition (32), and $(b)$ is by the convexity of both $V_{\boldsymbol{w}_i - \boldsymbol{w}_0}(\boldsymbol{u} - \boldsymbol{w}_0)$ and $\frac{1}{2\gamma} \|\boldsymbol{u} - \boldsymbol{w}_0\|^2$.

Meanwhile for $k \geq 1$, by the definition of $\psi_k(\boldsymbol{z}_k)$, we have

$$
\begin{aligned}
\psi_k(\boldsymbol{z}_k) &= \psi_{k-1}(\boldsymbol{z}_k) + a_k\langle F(\boldsymbol{w}_k), \boldsymbol{z}_k - \boldsymbol{u}\rangle \\
&\overset{(a)}{\geq} \psi_{k-1}(\boldsymbol{z}_{k-1}) + \frac{1+\sigma A_{k-1}}{2}\|\boldsymbol{z}_k - \boldsymbol{z}_{k-1}\|^2 + a_k\langle F(\boldsymbol{w}_k), \boldsymbol{z}_k - \boldsymbol{u}\rangle \\
&= \psi_{k-1}(\boldsymbol{z}_{k-1}) + \frac{1+\sigma A_{k-1}}{2}\|\boldsymbol{z}_k - \boldsymbol{z}_{k-1}\|^2 + a_k\langle F(\boldsymbol{w}_k), \boldsymbol{z}_k - \boldsymbol{w}_k\rangle \\
&\quad + a_k\langle F(\boldsymbol{w}_k), \boldsymbol{w}_k - \boldsymbol{u}\rangle,
\end{aligned}
\tag{34}
$$

where $(a)$ is by the $(1+\sigma A_{k-1})$-strong convexity of $\psi_{k-1}(\boldsymbol{z})$. Meanwhile, by the strong convexity of $\frac{1}{2}\|\cdot\|^2$ in Assumption 2, we have

$$
\begin{aligned}
&a_k\langle F(\boldsymbol{w}_k), \boldsymbol{w}_k - \boldsymbol{z}_k\rangle - \frac{1+\sigma A_{k-1}}{2}\|\boldsymbol{z}_k - \boldsymbol{z}_{k-1}\|^2 \\
&\leq \left\langle a_k F(\boldsymbol{w}_k) + (1+\sigma A_{k-1})\nabla_{\boldsymbol{w}_k}\frac{1}{2}\|\boldsymbol{w}_k - \boldsymbol{z}_{k-1}\|^2, \boldsymbol{w}_k - \boldsymbol{z}_k\right\rangle \\
&\quad - (1+\sigma A_{k-1})\Big(\frac{1}{2}\|\boldsymbol{w}_k - \boldsymbol{z}_{k-1}\|^2 + \frac{\gamma}{2}\|\boldsymbol{w}_k - \boldsymbol{z}_k\|^2\Big).
\end{aligned}
\tag{35}
$$

Then combining (34) and (35), and after simple arrangements, we have

$$
\begin{aligned}
a_k\langle F(\boldsymbol{w}_k), \boldsymbol{w}_k - \boldsymbol{u}\rangle &\leq \psi_k(\boldsymbol{z}_k) - \psi_{k-1}(\boldsymbol{z}_{k-1}) \\
&\quad + \left\langle a_k F(\boldsymbol{w}_k) + (1+\sigma A_{k-1})\nabla_{\boldsymbol{w}_k}\frac{1}{2}\|\boldsymbol{w}_k - \boldsymbol{z}_{k-1}\|^2, \boldsymbol{w}_k - \boldsymbol{z}_k\right\rangle \\
&\quad - (1+\sigma A_{k-1})\Big(\frac{1}{2}\|\boldsymbol{w}_k - \boldsymbol{z}_{k-1}\|^2 + \frac{\gamma}{2}\|\boldsymbol{w}_k - \boldsymbol{z}_k\|^2\Big).
\end{aligned}
\tag{36}
$$

Summing (36) from $k=1$ to $K$, we have

$$
\begin{aligned}
&\sum_{k=1}^{K} a_k\langle F(\boldsymbol{w}_k), \boldsymbol{w}_k - \boldsymbol{u}\rangle \\
&\leq \sum_{k=1}^{K}\left(\left\langle a_k F(\boldsymbol{w}_k) + (1+\sigma A_{k-1})\nabla_{\boldsymbol{w}_k}\frac{1}{2}\|\boldsymbol{w}_k - \boldsymbol{z}_{k-1}\|^2, \boldsymbol{w}_k - \boldsymbol{z}_k\right\rangle\right. \\
&\quad \left. - (1+\sigma A_{k-1})\Big(\frac{1}{2}\|\boldsymbol{w}_k - \boldsymbol{z}_{k-1}\|^2 + \frac{\gamma}{2}\|\boldsymbol{w}_k - \boldsymbol{z}_k\|^2\Big)\right) + \psi_K(\boldsymbol{z}_K) - \psi_0(\boldsymbol{z}_0) \\
&\overset{(a)}{\leq} \sum_{k=1}^{K}\left(\left\langle a_k F(\boldsymbol{w}_k) + (1+\sigma A_{k-1})\nabla_{\boldsymbol{w}_k}\frac{1}{2}\|\boldsymbol{w}_k - \boldsymbol{z}_{k-1}\|^2, \boldsymbol{w}_k - \boldsymbol{z}_k\right\rangle\right. \\
&\quad \left. - (1+\sigma A_{k-1})\Big(\frac{1}{2}\|\boldsymbol{w}_k - \boldsymbol{z}_{k-1}\|^2 + \frac{\gamma}{2}\|\boldsymbol{w}_k - \boldsymbol{z}_k\|^2\Big)\right) \\
&\quad + \frac{\sigma}{\gamma}\sum_{k=1}^{K} a_k V_{\boldsymbol{w}_k - \boldsymbol{w}_0}(\boldsymbol{u} - \boldsymbol{w}_0) + \frac{1}{2\gamma}\|\boldsymbol{u} - \boldsymbol{w}_0\|^2,
\end{aligned}
\tag{37}
$$

where $(a)$ is by the fact $\psi_0(\boldsymbol{z}_0) = 0$ and the upper bound of $\psi_K(\boldsymbol{z}_K)$ by (33). By the setting $a_k = \frac{\alpha\gamma(1+\sigma A_{k-1})}{L}$ in Algorithm 4 and (37), we have

$$
\begin{aligned}
&\sum_{k=1}^{K} a_k\langle F(\boldsymbol{w}_k), \boldsymbol{w}_k - \boldsymbol{u}\rangle \\
&\leq \sum_{k=1}^{K} a_k\left(\left\langle F(\boldsymbol{w}_k) + \frac{L}{\alpha\gamma}\nabla_{\boldsymbol{w}_k}\frac{1}{2}\|\boldsymbol{w}_k - \boldsymbol{z}_{k-1}\|^2, \boldsymbol{w}_k - \boldsymbol{z}_k\right\rangle\right. \\
&\quad \left. - \frac{L}{\alpha\gamma}\Big(\frac{1}{2}\|\boldsymbol{w}_k - \boldsymbol{z}_{k-1}\|^2 + \frac{\gamma}{2}\|\boldsymbol{w}_k - \boldsymbol{z}_k\|^2\Big)\right) \\
&\quad + \frac{\sigma}{\gamma}\sum_{k=1}^{K} a_k V_{\boldsymbol{w}_k - \boldsymbol{w}_0}(\boldsymbol{u} - \boldsymbol{w}_0) + \frac{1}{2\gamma}\|\boldsymbol{u} - \boldsymbol{w}_0\|^2.
\end{aligned}
\tag{38}
$$

Then based on the definition of $E_{1k}$ in Lemma 2, after simple arrangements, Lemma 2 is proved.

∎

## C.2   Proof of Lemma 3

*Proof.* By the definition of $E_{1k}$ in Lemma 2, we have: $\forall k \in [K]$,

$$
\begin{aligned}
E_{1k} &= a_k\Big(\Big\langle F(\boldsymbol{w}_k) + \frac{L}{\alpha\gamma}\nabla_{\boldsymbol{w}_k}\frac{1}{2}\|\boldsymbol{w}_k - \boldsymbol{z}_{k-1}\|^2, \boldsymbol{w}_k - \boldsymbol{z}_k\Big\rangle \\
&\quad - \frac{L}{\alpha\gamma}\Big(\frac{1}{2}\|\boldsymbol{w}_k - \boldsymbol{z}_{k-1}\|^2 + \frac{\gamma}{2}\|\boldsymbol{w}_k - \boldsymbol{z}_k\|^2\Big)\Big) \\
&\leq a_k\Big(\Big\langle F(\boldsymbol{w}_k) - F(\boldsymbol{w}_{k-1}), \boldsymbol{w}_k - \boldsymbol{z}_k\Big\rangle \\
&\quad + \Big\langle F(\boldsymbol{w}_{k-1}) + \frac{L}{\alpha\gamma}\nabla_{\boldsymbol{w}_k}\frac{1}{2}\|\boldsymbol{w}_k - \boldsymbol{z}_{k-1}\|^2, \boldsymbol{w}_k - \boldsymbol{z}_k\Big\rangle \\
&\quad - \frac{L}{\alpha\gamma}\Big(\frac{1}{2}\|\boldsymbol{w}_k - \boldsymbol{z}_{k-1}\|^2 + \frac{\gamma}{2}\|\boldsymbol{w}_k - \boldsymbol{z}_k\|^2\Big)\Big).
\end{aligned}
\tag{39}
$$

Meanwhile, we have for all $\alpha > 0$,

$$
\begin{aligned}
&\Big\langle F(\boldsymbol{w}_k) - F(\boldsymbol{w}_{k-1}), \boldsymbol{w}_k - \boldsymbol{z}_k\Big\rangle \\
&\overset{(a)}{\leq} \|F(\boldsymbol{w}_k) - F(\boldsymbol{w}_{k-1})\|_*\|\boldsymbol{w}_k - \boldsymbol{z}_k\| \\
&\overset{(b)}{\leq} L\|\boldsymbol{w}_k - \boldsymbol{w}_{k-1}\|\|\boldsymbol{w}_k - \boldsymbol{z}_k\| \\
&\overset{(c)}{\leq} L\alpha\|\boldsymbol{w}_k - \boldsymbol{w}_{k-1}\|^2 + \frac{L}{4\alpha}\|\boldsymbol{w}_k - \boldsymbol{z}_k\|^2 \\
&\overset{(d)}{\leq} L\alpha(\|\boldsymbol{w}_k - \boldsymbol{z}_{k-1}\| + \|\boldsymbol{z}_{k-1} - \boldsymbol{w}_{k-1}\|)^2 + \frac{L}{4\alpha}\|\boldsymbol{w}_k - \boldsymbol{z}_k\|^2 \\
&\overset{(e)}{\leq} 2L\alpha(\|\boldsymbol{w}_k - \boldsymbol{z}_{k-1}\|^2 + \|\boldsymbol{z}_{k-1} - \boldsymbol{w}_{k-1}\|^2) + \frac{L}{4\alpha}\|\boldsymbol{w}_k - \boldsymbol{z}_k\|^2,
\end{aligned}
\tag{40}
$$

where $(a)$ is by the Cauchy–Schwarz inequality, $(b)$ is the Lipschitz continuous Assumption 1, $(c)$ is by the fact $ab \leq a^2 + \frac{b^2}{4}$, $(d)$ is by the triangle inequality of norm $\|\cdot\|$ and $(e)$ is by the fact $(a+b)^2 \leq 2(a^2 + b^2)$.

Then by the optimality condition of $\boldsymbol{w}_k$ in the $k$-th iteration of Algorithm 4, we have: $\forall \boldsymbol{z} \in \mathcal{W}$,

$$
\Big\langle F(\boldsymbol{w}_{k-1}) + \frac{L}{\alpha\gamma}\nabla_{\boldsymbol{w}_k}\frac{1}{2}\|\boldsymbol{w}_k - \boldsymbol{z}_{k-1}\|^2, \boldsymbol{w}_k - \boldsymbol{z}\Big\rangle \leq 0.
\tag{41}
$$

By combining (39), (40) and (41), we have

$$
\begin{aligned}
E_{1k} &\leq a_k\Big(-\frac{L}{2\alpha\gamma}(1 - 4\alpha^2\gamma)\|\boldsymbol{w}_k - \boldsymbol{z}_{k-1}\|^2 + 2L\alpha\|\boldsymbol{w}_{k-1} - \boldsymbol{z}_{k-1}\|^2 - \frac{L}{4\alpha}\|\boldsymbol{w}_k - \boldsymbol{z}_k\|^2\Big) \\
&\overset{(a)}{\leq} a_k\Big(-\frac{L}{8\alpha\gamma}\|\boldsymbol{w}_k - \boldsymbol{z}_{k-1}\|^2 + \frac{L}{8\alpha}\frac{a_{k-1}}{a_k}\|\boldsymbol{w}_{k-1} - \boldsymbol{z}_{k-1}\|^2 - \frac{L}{4\alpha}\|\boldsymbol{w}_k - \boldsymbol{z}_k\|^2\Big) \\
&\leq -\frac{La_k}{8\alpha\gamma}\|\boldsymbol{w}_k - \boldsymbol{z}_{k-1}\|^2 + \frac{L}{8\alpha}(a_{k-1}\|\boldsymbol{w}_{k-1} - \boldsymbol{z}_{k-1}\|^2 - 2a_k\|\boldsymbol{w}_k - \boldsymbol{z}_k\|^2) \\
&\overset{(b)}{\leq} -\frac{La_k}{8\alpha}\|\boldsymbol{w}_k - \boldsymbol{z}_{k-1}\|^2 + \frac{L}{8\alpha}(a_{k-1}\|\boldsymbol{w}_{k-1} - \boldsymbol{z}_{k-1}\|^2 - 2a_k\|\boldsymbol{w}_k - \boldsymbol{z}_k\|^2),
\end{aligned}
\tag{42}
$$

where $(a)$ is by the fact

$$
a_k = \frac{\alpha\gamma}{L}\Big(1 + \frac{\alpha\gamma\sigma}{L}\Big)^{K-1}
\tag{43}
$$

from Step 5 of Algorithm 4 and the setting

$$
\alpha \leq \min\Big\{\frac{1}{4\sqrt{2}}, \frac{\sqrt{3}}{4\sqrt{\gamma}}\Big\} \leq \frac{1}{4}\sqrt{\frac{L}{L + \alpha\gamma\sigma}} = \frac{1}{4}\sqrt{\frac{a_{k-1}}{a_k}},
$$

and $(b)$ is by the setting that $0 < \gamma \leq 1$.

With the $\boldsymbol{w}_0 = \boldsymbol{z}_0$, for convenience, we set $a_0 := a_1$. By summing (42) from $k = 1$ to $K$, we have

$$
\begin{aligned}
\sum_{k=1}^{K} E_{1k} &\leq -\frac{L}{8\alpha}\Big(\sum_{k=1}^{K} a_k \big(\|\boldsymbol{w}_k - \boldsymbol{z}_{k-1}\|^2 + \|\boldsymbol{w}_k - \boldsymbol{z}_k\|^2\big) - a_0\|\boldsymbol{w}_0 - \boldsymbol{z}_0\|^2 + a_K\|\boldsymbol{w}_K - \boldsymbol{z}_K\|^2\Big) \\
&\overset{(a)}{=} -\frac{L}{8\alpha}\Big(\sum_{k=1}^{K} \big(a_k\|\boldsymbol{w}_k - \boldsymbol{z}_{k-1}\|^2 + a_{k-1}\|\boldsymbol{w}_{k-1} - \boldsymbol{z}_{k-1}\|^2\big) + 2a_K\|\boldsymbol{w}_K - \boldsymbol{z}_K\|^2\Big) \\
&\overset{(b)}{\leq} -\frac{L}{8\alpha}\sum_{k=1}^{K} a_{k-1}\big(\|\boldsymbol{w}_k - \boldsymbol{z}_{k-1}\|^2 + \|\boldsymbol{w}_{k-1} - \boldsymbol{z}_{k-1}\|^2\big),
\end{aligned}
\tag{44}
$$

where $(a)$ is by the fact $\boldsymbol{w}_0 = \boldsymbol{z}_0$, and $(b)$ is by the fact that $a_k \geq a_{k-1} > 0$. Lemma 3 is proved. ∎

### C.3 Proof of Lemma 4

*Proof.* It follows that

$$
\begin{aligned}
&\langle F(\boldsymbol{w}_k), \boldsymbol{w}_k - \boldsymbol{w}\rangle \\
&= \langle F(\boldsymbol{w}_k) - \big(F(\boldsymbol{w}_{k-1}) + \frac{L}{\alpha\gamma}\nabla_{\boldsymbol{w}_k}\frac{1}{2}\|\boldsymbol{w}_k - \boldsymbol{z}_{k-1}\|^2\big), \boldsymbol{w}_k - \boldsymbol{w}\rangle \\
&\quad + \Big\langle F(\boldsymbol{w}_{k-1}) + \frac{L}{\alpha\gamma}\nabla_{\boldsymbol{w}_k}\frac{1}{2}\|\boldsymbol{w}_k - \boldsymbol{z}_{k-1}\|^2, \boldsymbol{w}_k - \boldsymbol{w}\Big\rangle \\
&\overset{(a)}{\leq} \langle F(\boldsymbol{w}_k) - \big(F(\boldsymbol{w}_{k-1}) + \frac{L}{\alpha\gamma}\nabla_{\boldsymbol{w}_k}\frac{1}{2}\|\boldsymbol{w}_k - \boldsymbol{z}_{k-1}\|^2\big), \boldsymbol{w}_k - \boldsymbol{w}\rangle \\
&\overset{(b)}{\leq} \|F(\boldsymbol{w}_k) - F(\boldsymbol{w}_{k-1})\|_*\|\boldsymbol{w}_k - \boldsymbol{w}\| + \frac{L}{\alpha\gamma}\big\|\nabla_{\boldsymbol{w}_k}\frac{1}{2}\|\boldsymbol{w}_k - \boldsymbol{z}_{k-1}\|^2\big\|_*\|\boldsymbol{w}_k - \boldsymbol{w}\| \\
&\overset{(c)}{\leq} L\|\boldsymbol{w}_k - \boldsymbol{w}_{k-1}\|\|\boldsymbol{w}_k - \boldsymbol{w}\| + \frac{L}{\alpha\gamma}\delta\|\boldsymbol{w}_k - \boldsymbol{z}_{k-1}\|\|\boldsymbol{w}_k - \boldsymbol{w}\| \\
&\overset{(d)}{\leq} L(\|\boldsymbol{w}_k - \boldsymbol{z}_{k-1}\| + \|\boldsymbol{z}_{k-1} - \boldsymbol{w}_{k-1}\|)\|\boldsymbol{w}_k - \boldsymbol{w}\| + \frac{L}{\alpha\gamma}\delta\|\boldsymbol{w}_k - \boldsymbol{z}_{k-1}\|\|\boldsymbol{w}_k - \boldsymbol{w}\| \\
&\leq \Big(\big(1 + \frac{\delta}{\alpha\gamma}\big)(\|\boldsymbol{w}_k - \boldsymbol{z}_{k-1}\| + \|\boldsymbol{z}_{k-1} - \boldsymbol{w}_{k-1}\|)\Big)L\|\boldsymbol{w}_k - \boldsymbol{w}\|,
\end{aligned}
\tag{45}
$$

where $(a)$ is by the optimality condition of $\boldsymbol{w}_k$, $(b)$ is by the Cauchy-Schwarz inequality, $(c)$ is by the Lipschitz continuity of $F(\mathbf{w})$ and the bounded assumption (7), $(d)$ is by the triangle inequality of norm $\|\cdot\|$. So we have

$$
\sup_{\boldsymbol{w}\in\mathcal{W}, \|\boldsymbol{w}_k - \boldsymbol{w}\|\leq D} \langle F(\boldsymbol{w}_k), \boldsymbol{w}_k - \boldsymbol{w}\rangle \leq \big(1 + \frac{\delta}{\alpha\gamma}\big)DL(\|\boldsymbol{w}_k - \boldsymbol{z}_{k-1}\| + \|\boldsymbol{z}_{k-1} - \boldsymbol{w}_{k-1}\|).
\tag{46}
$$

Meanwhile, if there exists a $\boldsymbol{w}^*$ that satisfies Assumption 4, *i.e.*, $\forall \boldsymbol{w} \in \mathcal{W}$, $\langle F(\boldsymbol{w}), \boldsymbol{w} - \boldsymbol{w}^*\rangle \geq \frac{\sigma}{\gamma}(V_{\boldsymbol{w}-\boldsymbol{w}_0}(\boldsymbol{w}^* - \boldsymbol{w}_0) + V_{\boldsymbol{w}^*-\boldsymbol{w}_0}(\boldsymbol{w} - \boldsymbol{w}_0))$ with $\sigma > 0$, then in (45), let $\boldsymbol{w} := \boldsymbol{w}^*$, and by the fact $V_{\boldsymbol{w}_k-\boldsymbol{w}_0}(\boldsymbol{w}^* - \boldsymbol{w}_0) \geq \frac{\gamma}{2}\|\boldsymbol{w}_k - \boldsymbol{w}^*\|^2$ and $V_{\boldsymbol{w}^*-\boldsymbol{w}_0}(\boldsymbol{w}_k - \boldsymbol{w}_0) \geq \frac{\gamma}{2}\|\boldsymbol{w}_k - \boldsymbol{w}^*\|^2$, we have

$$
\begin{aligned}
\sigma\|\boldsymbol{w}_k - \boldsymbol{w}^*\|^2 &\leq \frac{\sigma}{\gamma}(V_{\boldsymbol{w}_k-\boldsymbol{w}_0}(\boldsymbol{w}^* - \boldsymbol{w}_0) + V_{\boldsymbol{w}^*-\boldsymbol{w}_0}(\boldsymbol{w}_k - \boldsymbol{w}_0)) \leq \langle F(\boldsymbol{w}_k), \boldsymbol{w}_k - \boldsymbol{w}^*\rangle \\
&\leq \big(1 + \frac{\delta}{\alpha\gamma}\big)L(\|\boldsymbol{w}_k - \boldsymbol{z}_{k-1}\| + \|\boldsymbol{z}_{k-1} - \boldsymbol{w}_{k-1}\|)\|\boldsymbol{w}_k - \boldsymbol{w}^*\|.
\end{aligned}
\tag{47}
$$

So it follows that

$$
\|\boldsymbol{w}_k - \boldsymbol{w}^*\| \leq \big(1 + \frac{\delta}{\alpha\gamma}\big)\frac{L}{\sigma}(\|\boldsymbol{w}_k - \boldsymbol{z}_{k-1}\| + \|\boldsymbol{z}_{k-1} - \boldsymbol{w}_{k-1}\|).
\tag{48}
$$

Lemma 4 is proved. ∎

### C.4 Proof of Theorem 1

*Proof.* Firstly, by the setting $a_k = \frac{\alpha\gamma(1+\sigma A_{k-1})}{L}$ and $A_0 = 0$, $A_k = A_{k-1} + a_k$, we have: $\forall k \geq 0$,

- If $\sigma = 0$, then $A_k = \frac{\alpha\gamma k}{L}$.

- If $\sigma > 0$, then $A_k = \frac{1}{\sigma}\left(1 + \frac{\alpha\gamma\sigma}{L}\right)^k - \frac{1}{\sigma}$.

By Lemmas 2 and 3, we have

$$\sum_{k=1}^{K} a_k \left(\langle F(\boldsymbol{w}_k), \boldsymbol{w}_k - \boldsymbol{u}\rangle - \frac{\sigma}{\gamma}V_{\boldsymbol{w}_k - \boldsymbol{w}_0}(\boldsymbol{u} - \boldsymbol{w}_0)\right)$$

$$\leq \sum_{k=1}^{K} E_{1k} + \frac{1}{2\gamma}\|\boldsymbol{u} - \boldsymbol{w}_0\|^2$$

$$\leq -\frac{L}{8\alpha}\sum_{k=1}^{K} a_{k-1}\big(\|\boldsymbol{w}_k - \boldsymbol{z}_{k-1}\|^2 + \|\boldsymbol{w}_{k-1} - \boldsymbol{z}_{k-1}\|^2\big) + \frac{1}{2\gamma}\|\boldsymbol{u} - \boldsymbol{w}_0\|^2. \tag{49}$$

Let $\boldsymbol{w}$ be the $\boldsymbol{w}^*$ in Assumption 3 if $\sigma = 0$ or the $\boldsymbol{w}^*$ in Assumption 4 if $\sigma > 0$. Then by the property of $\boldsymbol{w}^*$, we have $\langle F(\boldsymbol{w}_k), \boldsymbol{w}_k - \boldsymbol{w}^*\rangle - \frac{\sigma}{\gamma}V_{\boldsymbol{w}_k - \boldsymbol{w}_0}(\boldsymbol{w}^* - \boldsymbol{w}_0) \geq 0$. So by (49), it follows that

$$\frac{L}{16\alpha}\sum_{k=1}^{K} a_{k-1}\big(\|\boldsymbol{w}_k - \boldsymbol{z}_{k-1}\| + \|\boldsymbol{w}_{k-1} - \boldsymbol{z}_{k-1}\|\big)^2$$

$$\leq \frac{L}{8\alpha}\sum_{k=1}^{K} a_{k-1}\big(\|\boldsymbol{w}_k - \boldsymbol{z}_{k-1}\|^2 + \|\boldsymbol{w}_{k-1} - \boldsymbol{z}_{k-1}\|^2\big) \leq \frac{1}{2\gamma}\|\boldsymbol{w}^* - \boldsymbol{w}_0\|^2. \tag{50}$$

By the setting $A_k = A_{k-1} + a_k$ with $A_0 = 0$ in Algorithm 4, we have $A_k = \sum_{i=1}^{k} a_i$. Meanwhile, for convenience, we have set $a_0 = a_1$. So we have

$$\frac{L}{16\alpha}(A_{K-1} + a_1)\min_{k\in[K]}\big(\|\boldsymbol{w}_k - \boldsymbol{z}_{k-1}\| + \|\boldsymbol{w}_{k-1} - \boldsymbol{z}_{k-1}\|\big)^2 \leq \frac{1}{2\gamma}\|\boldsymbol{w}^* - \boldsymbol{w}_0\|^2. \tag{51}$$

So for the so computed $\{\boldsymbol{w}_k, \boldsymbol{z}_{k-1}\}$, let $\tilde{k} := \arg\min_{k\in[K]}(\|\boldsymbol{w}_k - \boldsymbol{z}_{k-1}\| + \|\boldsymbol{w}_{k-1} - \boldsymbol{z}_{k-1}\|)$ and $\tilde{\boldsymbol{w}}_K := \boldsymbol{w}_{\tilde{k}}$. Then combining (50) and (51), we have

$$\|\boldsymbol{w}_{\tilde{k}} - \boldsymbol{z}_{\tilde{k}-1}\| + \|\boldsymbol{w}_{\tilde{k}-1} - \boldsymbol{z}_{\tilde{k}-1}\| \leq \sqrt{\frac{\|\boldsymbol{w}^* - \boldsymbol{w}_0\|^2}{L(A_{K-1} + a_1)}\frac{8\alpha}{\gamma}}. \tag{52}$$

So by (20) of Lemma 4 and (52), it follows that

$$\sup_{\boldsymbol{w}\in\mathcal{W}, \|\tilde{\boldsymbol{w}}_K - \boldsymbol{w}\|\leq D} \langle F(\tilde{\boldsymbol{w}}_K), \tilde{\boldsymbol{w}}_K - \boldsymbol{w}\rangle$$

$$\leq \left(1 + \frac{\delta}{\alpha\gamma}\right)DL(\|\boldsymbol{w}_{\tilde{k}} - \boldsymbol{z}_{\tilde{k}-1}\| + \|\boldsymbol{w}_{\tilde{k}-1} - \boldsymbol{z}_{\tilde{k}-1}\|)$$

$$\leq \left(1 + \frac{\delta}{\alpha\gamma}\right)D\sqrt{\frac{L\|\boldsymbol{w}^* - \boldsymbol{w}_0\|^2}{(A_{K-1} + a_1)}\frac{8\alpha}{\gamma}}. \tag{53}$$

Similarly, if $\sigma > 0$, then by (21) of Lemma 4 and (52), we have

$$\|\tilde{\boldsymbol{w}}_K - \boldsymbol{w}^*\| \leq \left(1 + \frac{\delta}{\alpha\gamma}\right)\frac{L}{\sigma}(\|\boldsymbol{w}_{\tilde{k}} - \boldsymbol{z}_{\tilde{k}-1}\| + \|\boldsymbol{w}_{\tilde{k}-1} - \boldsymbol{z}_{\tilde{k}-1}\|)$$

$$\leq \left(1 + \frac{\delta}{\alpha\gamma}\right)\frac{1}{\sigma}\sqrt{\frac{L\|\boldsymbol{w}^* - \boldsymbol{w}_0\|^2}{(A_{K-1} + a_1)}\frac{8\alpha}{\gamma}}. \tag{54}$$

Then by defining $C_0 := \left(1 + \frac{\delta}{\alpha\gamma}\right)\sqrt{\frac{8\alpha}{\gamma}}$, Theorem 1 is proved.

∎

## C.5 Proof of Proposition 1

*Proof.* The proof follows the same paradigm of Section C.4. Firstly, by the setting $a_k = \frac{\alpha\gamma(1+\sigma A_{k-1})}{L}$ and $A_0 = 0$, $A_k = A_{k-1} + a_k$, we have $\forall k \geq 0$,

$$a_k = \frac{\alpha\gamma}{L}\left(1 + \frac{\alpha\gamma\sigma}{L}\right)^{k-1}. \tag{55}$$

Then the (51) of Section C.4 is replaced by

$$\frac{L}{16\alpha}a_{K-1}(\|\boldsymbol{w}_K - \boldsymbol{z}_{K-1}\| + \|\boldsymbol{w}_{K-1} - \boldsymbol{z}_{K-1}\|)^2 \leq \frac{1}{2\gamma}\|\boldsymbol{w}^* - \boldsymbol{w}_0\|^2. \tag{56}$$

Then similar to (52) to (54), we obtain the last iterate convergence result as

$$\sup_{\boldsymbol{w}\in\mathcal{W}, \|\tilde{\boldsymbol{w}}_K - \boldsymbol{w}\|\leq D} \langle F(\boldsymbol{w}_K), \boldsymbol{w}_K - \boldsymbol{w}\rangle \quad \leq \quad \left(1 + \frac{\delta}{\alpha\gamma}\right)D\sqrt{\frac{L\|\boldsymbol{w}^* - \boldsymbol{w}_0\|^2}{a_{K-1}}\frac{8\alpha}{\gamma}},$$

$$\|\boldsymbol{w}_K - \boldsymbol{w}^*\| \quad \leq \quad \left(1 + \frac{\delta}{\alpha\gamma}\right)\frac{1}{\sigma}\sqrt{\frac{L\|\boldsymbol{w}^* - \boldsymbol{w}_0\|^2}{a_{K-1}}\frac{8\alpha}{\gamma}}.$$

Thus by the definition of $C_0$ in Theorem 1, Proposition 1 is proved. ∎

## C.6 Proof of Lemma 1

*Proof.* By the definition of the Bregman divergence $V_{\boldsymbol{w}}(\boldsymbol{v})$, we have

$$V_{\boldsymbol{v}-\boldsymbol{w}_0}(\boldsymbol{w} - \boldsymbol{w}_0) = \frac{1}{2}\|\boldsymbol{w} - \boldsymbol{w}_0\|^2 - \frac{1}{2}\|\boldsymbol{v} - \boldsymbol{w}_0\|^2 - \langle\nabla_{\boldsymbol{v}}\frac{1}{2}\|\boldsymbol{v} - \boldsymbol{w}_0\|^2, \boldsymbol{w} - \boldsymbol{v}\rangle \tag{57}$$

$$V_{\boldsymbol{w}-\boldsymbol{w}_0}(\boldsymbol{v} - \boldsymbol{w}_0) = \frac{1}{2}\|\boldsymbol{v} - \boldsymbol{w}_0\|^2 - \frac{1}{2}\|\boldsymbol{w} - \boldsymbol{w}_0\|^2 - \langle\nabla_{\boldsymbol{w}}\frac{1}{2}\|\boldsymbol{w} - \boldsymbol{w}_0\|^2, \boldsymbol{v} - \boldsymbol{w}\rangle. \tag{58}$$

So combining (57) and (58), it follows that

$$\left\langle\nabla_{\boldsymbol{w}}\frac{1}{2}\|\boldsymbol{w} - \boldsymbol{w}_0\|^2 - \nabla_{\boldsymbol{v}}\frac{1}{2}\|\boldsymbol{v} - \boldsymbol{w}_0\|^2, \boldsymbol{w} - \boldsymbol{v}\right\rangle = V_{\boldsymbol{v}-\boldsymbol{w}_0}(\boldsymbol{w} - \boldsymbol{w}_0) + V_{\boldsymbol{w}-\boldsymbol{w}_0}(\boldsymbol{v} - \boldsymbol{w}_0). \tag{59}$$

So if $F(\boldsymbol{w})$ is monotone, then we have: $\forall \boldsymbol{w}_0, \boldsymbol{w}, \boldsymbol{v} \in \mathcal{W}$,

$$\left\langle\left(F(\boldsymbol{w}) + \epsilon\nabla_{\boldsymbol{w}}\frac{1}{2\gamma}\|\boldsymbol{w} - \boldsymbol{w}_0\|^2\right) - \left(F(\boldsymbol{v}) + \epsilon\nabla_{\boldsymbol{v}}\frac{1}{2\gamma}\|\boldsymbol{v} - \boldsymbol{w}_0\|^2\right), \boldsymbol{w} - \boldsymbol{v}\right\rangle$$

$$\geq \quad \frac{\epsilon}{\gamma}(V_{\boldsymbol{v}-\boldsymbol{w}_0}(\boldsymbol{w} - \boldsymbol{w}_0) + V_{\boldsymbol{w}-\boldsymbol{w}_0}(\boldsymbol{v} - \boldsymbol{w}_0)). \tag{60}$$

As Assumption 4 includes the strongly monotone assumption, by (60), we know that the VIP($F + \epsilon\nabla\frac{1}{2\gamma}\|\cdot - \boldsymbol{w}_0\|^2, \mathcal{W}$) satisfies Assumption 4 with parameter $\sigma = \epsilon$.

Lemma 1 is proved. ∎

## C.7 Proof of Corollary 1

*Proof.* By Theorem 1 and Lemma 1, if we optimize the regularized problem VIP($F + \epsilon\nabla\frac{1}{2\gamma}\|\cdot - \boldsymbol{w}_0\|^2, \mathcal{W}$) by the ODE Algorithm 4, then after $K$ iterations, we have

$$\sup_{\boldsymbol{w}\in\mathcal{W}, \|\tilde{\boldsymbol{w}}_K - \boldsymbol{w}\|\leq D} \langle F(\tilde{\boldsymbol{w}}_K) + \epsilon\nabla_{\tilde{\boldsymbol{w}}_K}\frac{1}{2\gamma}\|\tilde{\boldsymbol{w}}_K - \boldsymbol{w}_0\|^2, \tilde{\boldsymbol{w}}_K - \boldsymbol{w}\rangle$$

$$\leq C_0 D\|\boldsymbol{w}_0 - \boldsymbol{w}^*\|\sqrt{\frac{L}{A_{K-1} + a_1}}, \tag{61}$$

where $C_0$ is defined in Theorem 1, $A_{K-1} = \frac{1}{\epsilon}\left(1 + \frac{\sqrt{\alpha\gamma\epsilon}}{L}\right)^{K-1} - \frac{1}{\epsilon}$.

Meanwhile, by the convexity of $\frac{1}{2\gamma}\|\boldsymbol{w} - \boldsymbol{w}_0\|^2$, we have

$$\langle \nabla_{\tilde{\boldsymbol{w}}_K} \frac{1}{2\gamma}\|\tilde{\boldsymbol{w}}_K - \boldsymbol{w}_0\|^2, \boldsymbol{w} - \tilde{\boldsymbol{w}}_K \rangle \leq \frac{1}{2\gamma}\|\boldsymbol{w} - \boldsymbol{w}_0\|^2 - \frac{1}{2\gamma}\|\tilde{\boldsymbol{w}}_K - \boldsymbol{w}_0\|^2 \leq \frac{1}{2\gamma}\|\boldsymbol{w} - \boldsymbol{w}_0\|^2. \quad (62)$$

So combining (61) and (62), we have

$$\sup_{\boldsymbol{w}\in\mathcal{W}, \|\tilde{\boldsymbol{w}}_K - \boldsymbol{w}\|\leq D, \|\boldsymbol{w}-\boldsymbol{w}_0\|\leq D} \langle F(\tilde{\boldsymbol{w}}_K), \tilde{\boldsymbol{w}}_K - \boldsymbol{w} \rangle$$

$$\leq D\epsilon + DC_0\|\boldsymbol{w}_0 - \boldsymbol{w}^*\| \sqrt{\frac{L\epsilon}{\left(1 + \frac{\alpha\gamma\epsilon}{L}\right)^{K-1} - 1 + \frac{\alpha\gamma}{L}}}. \quad (63)$$

Corollary 1 is proved. ∎

### C.8    Proof of Corollary 2

*Proof.* By Proposition 1 and Lemma 1, if we optimize the regularized problem $\text{VIP}(F + \epsilon\nabla\frac{1}{2\gamma}\| \cdot -\boldsymbol{w}_0\|^2, \mathcal{W})$ by the OptDE Algorithm 4, then after $K$ iterations, we have

$$\sup_{\boldsymbol{w}\in\mathcal{W}, \|\boldsymbol{w}_K - \boldsymbol{w}\|\leq D} \langle F(\boldsymbol{w}_K) + \epsilon\nabla_{\boldsymbol{w}_K}\frac{1}{2\gamma}\|\boldsymbol{w}_K - \boldsymbol{w}_0\|^2, \boldsymbol{w}_K - \boldsymbol{w} \rangle$$

$$\leq C_0 D\|\boldsymbol{w}_0 - \boldsymbol{w}^*\| \sqrt{\frac{L}{a_{K-1}}}, \quad (64)$$

where $C_0$ is defined in Theorem 1, $a_{K-1} = \frac{\alpha\gamma}{L}\left(1 + \frac{\alpha\gamma\sigma}{L}\right)^{K-2}$.

Meanwhile, by the convexity of $\frac{1}{2\gamma}\|\boldsymbol{w} - \boldsymbol{w}_0\|^2$, we have

$$\langle \nabla_{\boldsymbol{w}_K} \frac{1}{2\gamma}\|\boldsymbol{w}_K - \boldsymbol{w}_0\|^2, \boldsymbol{w} - \boldsymbol{w}_K \rangle \leq \frac{1}{2\gamma}\|\boldsymbol{w} - \boldsymbol{w}_0\|^2 - \frac{1}{2\gamma}\|\boldsymbol{w}_K - \boldsymbol{w}_0\|^2 \leq \frac{1}{2\gamma}\|\boldsymbol{w} - \boldsymbol{w}_0\|^2. \quad (65)$$

So combining (64) and (65), we have

$$\sup_{\boldsymbol{w}\in\mathcal{W}, \|\boldsymbol{w}_K - \boldsymbol{w}\|\leq D, \|\boldsymbol{w}-\boldsymbol{w}_0\|\leq D} \langle F(\boldsymbol{w}_K), \boldsymbol{w}_K - \boldsymbol{w} \rangle$$

$$\leq D\epsilon + DC_0 L\|\boldsymbol{w}_0 - \boldsymbol{w}^*\| \sqrt{\frac{1}{\alpha\gamma\left(1 + \frac{\alpha\gamma\epsilon}{L}\right)^{K-2}}}.$$

Corollary 2 is proved. ∎

## D    Proof of Section B

By the definition of proximal operator (8), we can equivalently reformulate the stochastic optimistic dual extrapolation (SODE) of the main body as below. Then based on the definition of $\boldsymbol{g}_k$ in Step 7 and the definition of the Bregman divergence $V_{\boldsymbol{w}}(\boldsymbol{u})$, we can verify that

$$\boldsymbol{z}_k = \arg\min_{\boldsymbol{z}\in\mathcal{W}} \left\{ \hat{\psi}_k(\boldsymbol{z}) := \sum_{i=1}^k a_i \left( \langle F(\boldsymbol{w}_i; \xi_i), \boldsymbol{z} - \boldsymbol{u} \rangle + \frac{\sigma}{\gamma}V_{\boldsymbol{w}_i - \boldsymbol{w}_0}(\boldsymbol{z} - \boldsymbol{w}_0) \right) + \frac{1}{2\gamma}\|\boldsymbol{z} - \boldsymbol{w}_0\|^2 \right\}, \quad (66)$$

where $\boldsymbol{u}$ is an arbitrary vector in $\mathcal{W}$ and is irrelevant to the minimizer $\boldsymbol{z}_k$. In our context, $\hat{\psi}_k(\boldsymbol{z})$ plays the role of a "generalized estimation sequence" to help us conduct convergence analysis. By the $\gamma$-strong convexity of the Bregman divergence $V_{\boldsymbol{w}_i - \boldsymbol{w}_0}(\boldsymbol{z} - \boldsymbol{w}_0)$, we know that $\hat{\psi}_k(\boldsymbol{z})$ is strongly convex with strong convexity parameter $1 + \sigma \sum_{i=1}^k a_i = 1 + \sigma A_k$.

**Algorithm 5** Stochastic Optimistic Dual Extrapolation **(Reformulation)**

---

1: **Input:**   Lipshitz constant $L > 0$ from Assumption 1, $\gamma, \delta > 0$ from Assumption 2. The VIP$(F, \mathcal{W})$ satisfying Assumption 3 ($\sigma = 0$) or Assumption 4 ($\sigma > 0$).

2: $A_0 = 0, \alpha = \min\{\frac{\gamma}{32}, \frac{1}{16}\}$.

3: $\boldsymbol{w}_0 = \boldsymbol{z}_0 \in \mathcal{W}, \boldsymbol{g}_0 = \boldsymbol{0}$.

4: **for** $k = 1, 2, 3, \ldots, K$ **do**

5:  $\quad a_k = \frac{\alpha \gamma \sqrt{1 + \sigma A_{k-1}}}{L}, A_k = A_{k-1} + a_k$.

6:  $\quad \boldsymbol{w}_k = \arg\min_{\boldsymbol{w} \in \mathcal{W}} \left\{ \langle F(\boldsymbol{w}_{k-1}; \xi_{k-1}), \boldsymbol{w} \rangle + \frac{L^2 a_k}{2(\alpha \gamma)^2} \|\boldsymbol{w} - \boldsymbol{z}_{k-1}\|^2 \right\}$.

7:  $\quad \boldsymbol{g}_k = \boldsymbol{g}_{k-1} + a_k \left( F(\boldsymbol{w}_k; \xi_k) - \frac{\sigma}{\gamma} \nabla_{\boldsymbol{w}_k} \frac{1}{2} \|\boldsymbol{w}_k - \boldsymbol{w}_0\|^2 \right)$.

8:  $\quad \boldsymbol{z}_k = \arg\min_{\boldsymbol{z} \in \mathcal{W}} \left\{ \langle \boldsymbol{g}_k, \boldsymbol{z} \rangle + \frac{1 + \sigma A_k}{2\gamma} \|\boldsymbol{z} - \boldsymbol{w}_0\|^2 \right\}$

9: **end for**

10: $\tilde{\boldsymbol{w}}_K = \boldsymbol{w}_k$, where $k$ is chosen at random with probability distribution $\{\frac{a_1}{A_K}, \frac{a_2}{A_K}, \ldots, \frac{a_K}{A_K}\}$.

11: **return** $\tilde{\boldsymbol{w}}_K$.

---

### D.1   Proof of Lemma 5

*Proof.* Given the definition of the generalized estimation sequence $\hat{\psi}_k(\boldsymbol{z})$ in (66) and by the optimality condition of the minimizer $\boldsymbol{z}_k$ in the Step 6 of Algorithm 5, we have: $\forall \boldsymbol{u} \in \mathcal{W}$,

$$\left\langle \sum_{i=1}^{k} a_i (F(\boldsymbol{w}_i; \xi_i) + \frac{\sigma}{\gamma} \nabla V_{\boldsymbol{w}_i - \boldsymbol{w}_0}(\boldsymbol{z}_k - \boldsymbol{w}_0)) + \nabla_{\boldsymbol{z}_k} \frac{1}{2\gamma} \|\boldsymbol{z}_k - \boldsymbol{w}_0\|^2, \boldsymbol{u} - \boldsymbol{z}_k \right\rangle \geq 0. \qquad (67)$$

Then we have: $\forall k \in [K]$,

$$
\begin{aligned}
\hat{\psi}_k(\boldsymbol{z}_k) &= \sum_{i=1}^{k} a_i \left( \langle F(\boldsymbol{w}_i; \xi_i), \boldsymbol{z}_k - \boldsymbol{u} \rangle + \frac{\sigma}{\gamma} V_{\boldsymbol{w}_i - \boldsymbol{w}_0}(\boldsymbol{z}_k - \boldsymbol{w}_0) \right) + \frac{1}{2\gamma} \|\boldsymbol{z}_k - \boldsymbol{w}_0\|^2 \\
&\overset{(a)}{\leq} \frac{\sigma}{\gamma} \sum_{i=1}^{k} a_i \left( \langle \nabla V_{\boldsymbol{w}_i - \boldsymbol{w}_0}(\boldsymbol{z}_k - \boldsymbol{w}_0), \boldsymbol{u} - \boldsymbol{z}_k \rangle + V_{\boldsymbol{w}_i - \boldsymbol{w}_0}(\boldsymbol{z}_k - \boldsymbol{w}_0) \right) \\
&\quad + \left\langle \nabla_{\boldsymbol{z}_k} \frac{1}{2\gamma} \|\boldsymbol{z}_k - \boldsymbol{w}_0\|^2, \boldsymbol{u} - \boldsymbol{z}_k \right\rangle + \frac{1}{2\gamma} \|\boldsymbol{z}_k - \boldsymbol{w}_0\|^2 \\
&\overset{(b)}{\leq} \frac{\sigma}{\gamma} \sum_{i=1}^{k} a_i V_{\boldsymbol{w}_i - \boldsymbol{w}_0}(\boldsymbol{u} - \boldsymbol{w}_0) + \frac{1}{2\gamma} \|\boldsymbol{u} - \boldsymbol{w}_0\|^2, \qquad (68)
\end{aligned}
$$

where $(a)$ is by the optimality condition (67) and $(b)$ is by the convexity of $V_{\boldsymbol{w}_i - \boldsymbol{w}_0}(\boldsymbol{u} - \boldsymbol{w}_0)$ and $\frac{1}{2\gamma} \|\boldsymbol{u} - \boldsymbol{w}_0\|^2$.

Meanwhile $\forall k \in [K]$, we have

$$
\begin{aligned}
\hat{\psi}_k(\boldsymbol{z}_k) &= \hat{\psi}_{k-1}(\boldsymbol{z}_k) + a_k \langle F(\boldsymbol{w}_k; \xi_k), \boldsymbol{z}_k - \boldsymbol{u} \rangle \\
&\overset{(a)}{\geq} \hat{\psi}_{k-1}(\boldsymbol{z}_{k-1}) + \frac{1 + \sigma A_{k-1}}{2} \|\boldsymbol{z}_k - \boldsymbol{z}_{k-1}\|^2 + a_k \langle F(\boldsymbol{w}_k; \xi_k), \boldsymbol{z}_k - \boldsymbol{u} \rangle \\
&= \hat{\psi}_{k-1}(\boldsymbol{z}_{k-1}) + \frac{1 + \sigma A_{k-1}}{2} \|\boldsymbol{z}_k - \boldsymbol{z}_{k-1}\|^2 \\
&\quad + a_k \langle F(\boldsymbol{w}_k; \xi_k), \boldsymbol{z}_k - \boldsymbol{w}_k \rangle + a_k \langle F(\boldsymbol{w}_k; \xi_k), \boldsymbol{w}_k - \boldsymbol{u} \rangle, \qquad (69)
\end{aligned}
$$

where $(a)$ is the $(1 + \sigma A_{k-1})$-strong convexity of $\hat{\psi}_{k-1}(\boldsymbol{z})$. Meanwhile, by the $\gamma$-strong convexity of $\frac{1}{2} \| \cdot \|^2$, we have

$$
\begin{aligned}
&a_k \langle F(\boldsymbol{w}_k; \xi_k), \boldsymbol{w}_k - \boldsymbol{z}_k \rangle - \frac{1 + \sigma A_{k-1}}{2} \|\boldsymbol{z}_k - \boldsymbol{z}_{k-1}\|^2 \\
&\leq \left\langle a_k F(\boldsymbol{w}_k; \xi_k) + (1 + \sigma A_{k-1}) \nabla_{\boldsymbol{w}_k} \frac{1}{2} \|\boldsymbol{w}_k - \boldsymbol{z}_{k-1}\|^2, \boldsymbol{w}_k - \boldsymbol{z}_k \right\rangle \\
&\quad - (1 + \sigma A_{k-1}) \left( \frac{1}{2} \|\boldsymbol{w}_k - \boldsymbol{z}_{k-1}\|^2 + \frac{\gamma}{2} \|\boldsymbol{w}_k - \boldsymbol{z}_k\|^2 \right). \qquad (70)
\end{aligned}
$$

Then combining (69) and (70), we have

$$a_k\langle F(\boldsymbol{w}_k;\xi_k), \boldsymbol{w}_k - \boldsymbol{u}\rangle$$

$$\leq \quad \Big\langle a_k F(\boldsymbol{w}_k;\xi_k) + (1+\sigma A_{k-1})\nabla_{\boldsymbol{w}_k}\frac{1}{2}\|\boldsymbol{w}_k - \boldsymbol{z}_{k-1}\|^2, \boldsymbol{w}_k - \boldsymbol{z}_k\Big\rangle$$

$$-(1+\sigma A_{k-1})\Big(\frac{1}{2}\|\boldsymbol{w}_k - \boldsymbol{z}_{k-1}\|^2 + \frac{\gamma}{2}\|\boldsymbol{w}_k - \boldsymbol{z}_k\|^2\Big) + \hat{\psi}_k(\boldsymbol{z}_k) - \hat{\psi}_{k-1}(\boldsymbol{z}_{k-1}). \quad (71)$$

Summing (71) from $k = 1$ to $K$, we have

$$\sum_{k=1}^{K} a_k\langle F(\boldsymbol{w}_k;\xi_k), \boldsymbol{w}_k - \boldsymbol{u}\rangle$$

$$\leq \quad \sum_{k=1}^{K}\Big(\Big\langle a_k F(\boldsymbol{w}_k;\xi_k) + (1+\sigma A_{k-1})\nabla_{\boldsymbol{w}_k}\frac{1}{2}\|\boldsymbol{w}_k - \boldsymbol{z}_{k-1}\|^2, \boldsymbol{w}_k - \boldsymbol{z}_k\Big\rangle$$

$$-(1+\sigma A_{k-1})\Big(\frac{1}{2}\|\boldsymbol{w}_k - \boldsymbol{z}_{k-1}\|^2 + \frac{\gamma}{2}\|\boldsymbol{w}_k - \boldsymbol{z}_k\|^2\Big)\Big) + \hat{\psi}_K(\boldsymbol{z}_K) - \hat{\psi}_0(\boldsymbol{z}_0)$$

$$\overset{(a)}{\leq} \quad \sum_{k=1}^{K}\Big(\Big\langle a_k F(\boldsymbol{w}_k;\xi_k) + (1+\sigma A_{k-1})\nabla_{\boldsymbol{w}_k}\frac{1}{2}\|\boldsymbol{w}_k - \boldsymbol{z}_{k-1}\|^2, \boldsymbol{w}_k - \boldsymbol{z}_k\Big\rangle$$

$$-(1+\sigma A_{k-1})\Big(\frac{1}{2}\|\boldsymbol{w}_k - \boldsymbol{z}_{k-1}\|^2 + \frac{\gamma}{2}\|\boldsymbol{w}_k - \boldsymbol{z}_k\|^2\Big)\Big)$$

$$+\frac{\sigma}{\gamma}\sum_{k=1}^{K} a_k V_{\boldsymbol{w}_k - \boldsymbol{w}_0}(\boldsymbol{u} - \boldsymbol{w}_0) + \frac{1}{2\gamma}\|\boldsymbol{u} - \boldsymbol{w}_0\|^2$$

$$\overset{(b)}{=} \quad \sum_{k=1}^{K} a_k\Big(\Big\langle F(\boldsymbol{w}_k;\xi_k) + \frac{L^2 a_k}{(\alpha\gamma)^2}\nabla_{\boldsymbol{w}_k}\frac{1}{2}\|\boldsymbol{w}_k - \boldsymbol{z}_{k-1}\|^2, \boldsymbol{w}_k - \boldsymbol{z}_k\Big\rangle$$

$$-\frac{L^2 a_k}{(\alpha\gamma)^2}\Big(\frac{1}{2}\|\boldsymbol{w}_k - \boldsymbol{z}_{k-1}\|^2 + \frac{\gamma}{2}\|\boldsymbol{w}_k - \boldsymbol{z}_k\|^2\Big)\Big)$$

$$+\frac{\sigma}{\gamma}\sum_{k=1}^{K} a_k V_{\boldsymbol{w}_k - \boldsymbol{w}_0}(\boldsymbol{u} - \boldsymbol{w}_0) + \frac{1}{2\gamma}\|\boldsymbol{u} - \boldsymbol{w}_0\|^2, \quad (72)$$

where $(a)$ is by the fact $\hat{\psi}_0(\boldsymbol{z}_0) = 0$, the upper bound of $\hat{\psi}_K(\boldsymbol{z}_K)$ in (68), $(b)$ is by the setting $a_k^2 = \frac{(\alpha\gamma)^2(1+\sigma A_{k-1})}{L^2}$ in Algorithm 5. Meanwhile, taking expectation on $\xi_k$, we have: $\forall \boldsymbol{u} \in \mathcal{W}$,

$$\langle F(\boldsymbol{w}_k), \boldsymbol{w}_k - \boldsymbol{u}\rangle \quad = \quad \mathbb{E}_{\xi_k}\Big[\langle F(\boldsymbol{w}_k) - F(\boldsymbol{w}_k;\xi_k), \boldsymbol{w}_k - \boldsymbol{u}\rangle\Big] + \mathbb{E}_{\xi_k}\Big[\langle F(\boldsymbol{w}_k;\xi_k), \boldsymbol{w}_k - \boldsymbol{u}\rangle\Big]$$

$$= \quad \mathbb{E}_{\xi_k}\Big[\langle F(\boldsymbol{w}_k;\xi_k), \boldsymbol{w}_k - \boldsymbol{u}\rangle\Big]. \quad (73)$$

So taking expectation on the randomness of all the history for (72), and using (73) and the definition of $\{E_{2k}\}$ in Lemma 5, after simple arrangements, Lemma 5 is proved. ∎

### D.2 Proof of Lemma 6

*Proof.* By the definition of $E_{2k}$ in Lemma 5, we have: $\forall k \in [K]$,

$$
\begin{aligned}
E_{2k} &= a_k\Big(\Big\langle F(\boldsymbol{w}_k;\xi_k) + \frac{L^2 a_k}{(\alpha\gamma)^2}\nabla_{\boldsymbol{w}_k}\frac{1}{2}\|\boldsymbol{w}_k - \boldsymbol{z}_{k-1}\|^2, \boldsymbol{w}_k - \boldsymbol{z}_k\Big\rangle \\
&\quad -\frac{L^2 a_k}{(\alpha\gamma)^2}\Big(\frac{1}{2}\|\boldsymbol{w}_k - \boldsymbol{z}_{k-1}\|^2 + \frac{\gamma}{2}\|\boldsymbol{w}_k - \boldsymbol{z}_k\|^2\Big)\Big) \\
&\quad + a_k\langle F(\boldsymbol{w}_k) - F(\boldsymbol{w}_k;\xi_k), \boldsymbol{w}_k - \boldsymbol{u}\rangle \\
&\leq a_k\Big(\Big\langle F(\boldsymbol{w}_k;\xi_k) - F(\boldsymbol{w}_{k-1};\xi_{k-1}), \boldsymbol{w}_k - \boldsymbol{z}_k\Big\rangle \\
&\quad +\Big\langle F(\boldsymbol{w}_{k-1};\xi_{k-1}) + \frac{L^2 a_k}{(\alpha\gamma)^2}\nabla_{\boldsymbol{w}_k}\frac{1}{2}\|\boldsymbol{w}_k - \boldsymbol{z}_{k-1}\|^2, \boldsymbol{w}_k - \boldsymbol{z}_k\Big\rangle \\
&\quad -\frac{L^2 a_k}{(\alpha\gamma)^2}\Big(\frac{1}{2}\|\boldsymbol{w}_k - \boldsymbol{z}_{k-1}\|^2 + \frac{\gamma}{2}\|\boldsymbol{w}_k - \boldsymbol{z}_k\|^2\Big)\Big) \\
&\quad + a_k\langle F(\boldsymbol{w}_k) - F(\boldsymbol{w}_k;\xi_k), \boldsymbol{w}_k - \boldsymbol{u}\rangle. \tag{74}
\end{aligned}
$$

Meanwhile, we have: for all $\alpha > 0$,

$$
\Big\langle F(\boldsymbol{w}_k;\xi_k) - F(\boldsymbol{w}_{k-1};\xi_{k-1}), \boldsymbol{w}_k - \boldsymbol{z}_k\Big\rangle
$$

$$
\overset{(a)}{\leq} \|F(\boldsymbol{w}_k;\xi_k) - F(\boldsymbol{w}_{k-1};\xi_{k-1})\|_*\|\boldsymbol{w}_k - \boldsymbol{z}_k\|
$$

$$
\overset{(b)}{\leq} (\|F(\boldsymbol{w}_k) - F(\boldsymbol{w}_{k-1})\|_* + \|F(\boldsymbol{w}_k) - F(\boldsymbol{w}_k;\xi_k)\|_* \\
\quad + \|F(\boldsymbol{w}_{k-1}) - F(\boldsymbol{w}_{k-1};\xi_{k-1})\|_*)\|\boldsymbol{w}_k - \boldsymbol{z}_k\|
$$

$$
\overset{(c)}{\leq} (L\|\boldsymbol{w}_k - \boldsymbol{w}_{k-1}\| + \|F(\boldsymbol{w}_k) - F(\boldsymbol{w}_k;\xi_k)\|_* + \|F(\boldsymbol{w}_{k-1}) - F(\boldsymbol{w}_{k-1};\xi_{k-1})\|_*)\|\boldsymbol{w}_k - \boldsymbol{z}_k\|
$$

$$
\overset{(d)}{\leq} \frac{\alpha}{L^2 a_k}(L\|\boldsymbol{w}_k - \boldsymbol{w}_{k-1}\| + \|F(\boldsymbol{w}_k) - F(\boldsymbol{w}_k;\xi_k)\|_* + \|F(\boldsymbol{w}_{k-1}) - F(\boldsymbol{w}_{k-1};\xi_{k-1})\|_*)^2 \\
\quad + \frac{L^2 a_k}{4\alpha}\|\boldsymbol{w}_k - \boldsymbol{z}_k\|^2.
$$

$$
\overset{(e)}{\leq} \frac{2\alpha}{a_k}\|\boldsymbol{w}_k - \boldsymbol{w}_{k-1}\|^2 + \frac{2\alpha}{L^2 a_k}(\|F(\boldsymbol{w}_k) - F(\boldsymbol{w}_k;\xi_k)\|_* + \|F(\boldsymbol{w}_{k-1}) - F(\boldsymbol{w}_{k-1};\xi_k)\|_*)^2 \\
\quad + \frac{L^2 a_k}{4\alpha}\|\boldsymbol{w}_k - \boldsymbol{z}_k\|^2
$$

$$
\overset{(f)}{\leq} \frac{2\alpha}{a_k}\|\boldsymbol{w}_k - \boldsymbol{w}_{k-1}\|^2 + \frac{4\alpha}{L^2 a_k}(\|F(\boldsymbol{w}_k) - F(\boldsymbol{w}_k;\xi_k)\|_*^2 + \|F(\boldsymbol{w}_{k-1}) - F(\boldsymbol{w}_{k-1};\xi_k)\|_*^2) \\
\quad + \frac{L^2 a_k}{4\alpha}\|\boldsymbol{w}_k - \boldsymbol{z}_k\|^2
$$

$$
\overset{(g)}{\leq} \frac{4\alpha}{a_k}(\|\boldsymbol{w}_k - \boldsymbol{z}_{k-1}\|^2 + \|\boldsymbol{z}_{k-1} - \boldsymbol{w}_{k-1}\|^2) \\
\quad + \frac{4\alpha}{L^2 a_k}(\|F(\boldsymbol{w}_k) - F(\boldsymbol{w}_k;\xi_k)\|_*^2 + \|F(\boldsymbol{w}_{k-1}) - F(\boldsymbol{w}_{k-1};\xi_k)\|_*^2) + \frac{L^2 a_k}{4\alpha}\|\boldsymbol{w}_k - \boldsymbol{z}_k\|^2, \tag{75}
$$

where $(a)$ is by the Cauchy-Schwarz inequality, $(b)$ is by the triangle inequality of the norm $\|\cdot\|_*$, $(c)$ is by the Lipschitz continuity of $F(\boldsymbol{w})$, $(d)$ is by the fact $ab \leq a^2 + \frac{b^2}{4}$, $(e), (f)$ and $(g)$ is by the fact $(a+b)^2 \leq 2(a^2 + b^2)$.

Then by the optimality condition of $\boldsymbol{w}_k$ in Algorithm 5, we have: $\forall \boldsymbol{z} \in \mathcal{W}$,

$$
\Big\langle F(\boldsymbol{w}_{k-1};\xi_{k-1}) + \frac{L^2 a_k}{(\alpha\gamma)^2}\nabla_{\boldsymbol{w}_k}\frac{1}{2}\|\boldsymbol{w}_k - \boldsymbol{z}_{k-1}\|^2, \boldsymbol{w}_k - \boldsymbol{z}\Big\rangle \leq 0. \tag{76}
$$

Combining (74), (75) and (76) with $\boldsymbol{z} := \boldsymbol{z}_k$, we have

$$
\begin{aligned}
E_{2k} \leq\ & -\Big(\frac{L^2 a_k^2}{2(\alpha\gamma)^2} - 4\alpha\Big)\|\boldsymbol{w}_k - \boldsymbol{z}_{k-1}\|^2 \\
& + \frac{4\alpha}{L^2}\big(\|F(\boldsymbol{w}_k) - F(\boldsymbol{w}_k;\xi_k)\|_*^2 + \|F(\boldsymbol{w}_{k-1}) - F(\boldsymbol{w}_{k-1};\xi_{k-1})\|_*^2\big) \\
& + 4\alpha\|\boldsymbol{z}_{k-1} - \boldsymbol{w}_{k-1}\|^2 - \frac{L^2 a_k^2}{4\alpha}\|\boldsymbol{w}_k - \boldsymbol{z}_k\|^2 + a_k\langle F(\boldsymbol{w}_k) - F(\boldsymbol{w}_k;\xi_k), \boldsymbol{w}_k - \boldsymbol{u}\rangle.
\end{aligned}
$$

For both the settings $\sigma = 0$ and $\sigma > 0$, by our setting, we have $a_k \geq a_1 = \frac{\alpha\gamma}{L}$ and $\alpha = \min\{\frac{\gamma}{32}, \frac{1}{16}\}$, so we have

$$
\frac{L^2 a_k^2}{2(\alpha\gamma)^2} \geq \frac{1}{2} \geq 8\alpha, \qquad \frac{L^2 a_k^2}{4\alpha} \geq 8\alpha. \tag{77}
$$

Then it follows that

$$
\begin{aligned}
E_{2k} \leq\ & -4\alpha\|\boldsymbol{w}_k - \boldsymbol{z}_{k-1}\|^2 + \frac{4\alpha}{L^2}\big(\|F(\boldsymbol{w}_k) - F(\boldsymbol{w}_k;\xi_k)\|_*^2 + \|F(\boldsymbol{w}_{k-1}) - F(\boldsymbol{w}_{k-1};\xi_{k-1})\|_*^2\big) \\
& + 4\alpha\|\boldsymbol{z}_{k-1} - \boldsymbol{w}_{k-1}\|^2 - 8\alpha\|\boldsymbol{w}_k - \boldsymbol{z}_k\|^2 + \frac{\alpha\gamma}{L}\langle F(\boldsymbol{w}_k) - F(\boldsymbol{w}_k;\xi_k), \boldsymbol{w}_k - \boldsymbol{u}\rangle. \tag{78}
\end{aligned}
$$

So summing (78) from $k = 1$ to $K$ and by the fact $\mathbb{E}[\langle F(\boldsymbol{w}_k) - F(\boldsymbol{w}_k;\xi_k), \boldsymbol{w}_k - \boldsymbol{u}\rangle] = 0$, we have

$$
\begin{aligned}
\mathbb{E}\Big[\sum_{k=1}^{K} E_{2k}\Big] \leq\ & \mathbb{E}\Big[-4\alpha\sum_{k=1}^{K}\big(\|\boldsymbol{w}_k - \boldsymbol{z}_{k-1}\|^2 + \|\boldsymbol{w}_k - \boldsymbol{z}_k\|^2\big) \\
& + \frac{4\alpha}{L^2}\sum_{k=1}^{K}\big(\|F(\boldsymbol{w}_k) - F(\boldsymbol{w}_k;\xi_k)\|_*^2 + \|F(\boldsymbol{w}_{k-1}) - F(\boldsymbol{w}_{k-1};\xi_{k-1})\|_*^2\big)\Big] \\
\overset{(a)}{\leq}\ & -\mathbb{E}\Big[4\alpha\sum_{k=1}^{K}\big(\|\boldsymbol{w}_k - \boldsymbol{z}_{k-1}\|^2 + \|\boldsymbol{w}_{k-1} - \boldsymbol{z}_{k-1}\|^2\big)\Big] - \mathbb{E}[4\alpha\|\boldsymbol{w}_K - \boldsymbol{z}_K\|^2] + \frac{8\alpha s^2 K}{L^2} \\
\leq\ & -\mathbb{E}\Big[4\alpha\sum_{k=1}^{K}\big(\|\boldsymbol{w}_k - \boldsymbol{z}_{k-1}\|^2 + \|\boldsymbol{w}_{k-1} - \boldsymbol{z}_{k-1}\|^2\big)\Big] + \frac{8\alpha s^2 K}{L^2},
\end{aligned}
\tag{79}
$$

where $(a)$ is by the condition $\boldsymbol{w}_0 = \boldsymbol{z}_0$ and Assumption 5. Lemma 6 is proved.

∎

### D.3  Proof of Lemma 7

It follows that: $\forall \boldsymbol{w} \in \mathcal{W}$

$$
\begin{aligned}
& \langle F(\boldsymbol{w}_k), \boldsymbol{w}_k - \boldsymbol{w}\rangle \\
=\ & \Big\langle F(\boldsymbol{w}_k) - \Big(F(\boldsymbol{w}_{k-1};\xi_{k-1}) + \frac{L^2 a_k}{(\alpha\gamma)^2}\nabla_{\boldsymbol{w}_k}\frac{1}{2}\|\boldsymbol{w}_k - \boldsymbol{z}_{k-1}\|^2\Big), \boldsymbol{w}_k - \boldsymbol{w}\Big\rangle \\
& + \Big\langle F(\boldsymbol{w}_{k-1};\xi_{k-1}) + \frac{L^2 a_k}{(\alpha\gamma)^2}\nabla_{\boldsymbol{w}_k}\frac{1}{2}\|\boldsymbol{w}_k - \boldsymbol{z}_{k-1}\|^2, \boldsymbol{w}_k - \boldsymbol{w}\Big\rangle \\
\overset{(a)}{=}\ & \Big\langle F(\boldsymbol{w}_k) - \Big(F(\boldsymbol{w}_{k-1};\xi_{k-1}) + \frac{L}{\alpha\gamma}\nabla_{\boldsymbol{w}_k}\frac{1}{2}\|\boldsymbol{w}_k - \boldsymbol{z}_{k-1}\|^2\Big), \boldsymbol{w}_k - \boldsymbol{w}\Big\rangle \\
& + \Big\langle F(\boldsymbol{w}_{k-1};\xi_{k-1}) + \frac{L}{\alpha\gamma}\nabla_{\boldsymbol{w}_k}\frac{1}{2}\|\boldsymbol{w}_k - \boldsymbol{z}_{k-1}\|^2, \boldsymbol{w}_k - \boldsymbol{w}\Big\rangle \\
\overset{(b)}{\leq}\ & \Big\langle F(\boldsymbol{w}_k) - \Big(F(\boldsymbol{w}_{k-1};\xi_{k-1}) + \frac{L}{\alpha\gamma}\nabla_{\boldsymbol{w}_k}\frac{1}{2}\|\boldsymbol{w}_k - \boldsymbol{z}_{k-1}\|^2\Big), \boldsymbol{w}_k - \boldsymbol{w}\Big\rangle \\
\leq\ & \langle F(\boldsymbol{w}_k) - F(\boldsymbol{w}_{k-1}), \boldsymbol{w}_k - \boldsymbol{w}\rangle + \langle F(\boldsymbol{w}_{k-1}) - F(\boldsymbol{w}_{k-1};\xi_{k-1}), \boldsymbol{w}_k - \boldsymbol{w}\rangle \\
& + \Big\langle \frac{L}{\alpha\gamma}\nabla_{\boldsymbol{w}_k}\frac{1}{2}\|\boldsymbol{w}_k - \boldsymbol{z}_{k-1}\|^2, \boldsymbol{w}_k - \boldsymbol{w}\Big\rangle,
\end{aligned}
$$

where $(a)$ is by the fact $a_k = \frac{\alpha\gamma}{L}$ when $\sigma = 0$, $(b)$ is by the optimality condition of $\boldsymbol{w}_k$.

So it follows that

$$
\begin{aligned}
&\langle F(\boldsymbol{w}_k), \boldsymbol{w}_k - \boldsymbol{w}\rangle \\
&\stackrel{(a)}{\leq} \|F(\boldsymbol{w}_k) - F(\boldsymbol{w}_{k-1})\|_* \|\boldsymbol{w}_k - \boldsymbol{w}\| \\
&\quad + \langle F(\boldsymbol{w}_{k-1}) - F(\boldsymbol{w}_{k-1}; \xi_{k-1}), \boldsymbol{w}_k - \boldsymbol{w}_{k-1}\rangle \\
&\quad + \langle F(\boldsymbol{w}_{k-1}) - F(\boldsymbol{w}_{k-1}; \xi_{k-1}), \boldsymbol{w}_{k-1} - \boldsymbol{w}\rangle \\
&\quad + \frac{L}{\alpha\gamma} \big\|\nabla_{\boldsymbol{w}_k} \tfrac{1}{2}\|\boldsymbol{w}_k - \boldsymbol{z}_{k-1}\|^2\big\|_* \|\boldsymbol{w}_k - \boldsymbol{w}\| \\
&\stackrel{(b)}{\leq} L\|\boldsymbol{w}_k - \boldsymbol{w}_{k-1}\|\|\boldsymbol{w}_k - \boldsymbol{w}\| + \|F(\boldsymbol{w}_{k-1}) - F(\boldsymbol{w}_{k-1}; \xi_{k-1})\|_* \|\boldsymbol{w}_k - \boldsymbol{w}_{k-1}\| \\
&\quad + \langle F(\boldsymbol{w}_{k-1}) - F(\boldsymbol{w}_{k-1}; \xi_{k-1}), \boldsymbol{w}_{k-1} - \boldsymbol{w}\rangle + \frac{L\delta}{\alpha\gamma}\|\boldsymbol{w}_k - \boldsymbol{z}_{k-1}\|\|\boldsymbol{w}_k - \boldsymbol{w}\| \\
&\leq \big(1 + \frac{\delta}{\alpha\gamma}\big)L(\|\boldsymbol{w}_k - \boldsymbol{z}_{k-1}\| + \|\boldsymbol{w}_{k-1} - \boldsymbol{z}_{k-1}\|)\|\boldsymbol{w}_k - \boldsymbol{w}\| \\
&\quad + \|F(\boldsymbol{w}_{k-1}) - F(\boldsymbol{w}_{k-1}; \xi_{k-1})\|_* \|\boldsymbol{w}_k - \boldsymbol{w}_{k-1}\| \\
&\quad + \langle F(\boldsymbol{w}_{k-1}) - F(\boldsymbol{w}_{k-1}; \xi_{k-1}), \boldsymbol{w}_{k-1} - \boldsymbol{w}\rangle,
\end{aligned} \tag{80}
$$

$(a)$ is by the Cauchy Schwarz inequality and simple arrangement, and $(b)$ is by Assumption 1.

Then

$$
\begin{aligned}
&\langle F(\boldsymbol{w}_k), \boldsymbol{w}_k - \boldsymbol{w}\rangle \\
&\stackrel{(a)}{\leq} \big(1 + \frac{\delta}{\alpha\gamma}\big)L(\|\boldsymbol{w}_k - \boldsymbol{z}_{k-1}\| + \|\boldsymbol{w}_{k-1} - \boldsymbol{z}_{k-1}\|)\|\boldsymbol{w}_k - \boldsymbol{w}\| \\
&\quad + \frac{1}{2L^2}\|F(\boldsymbol{w}_{k-1}) - F(\boldsymbol{w}_{k-1}; \xi_{k-1})\|_*^2 \\
&\quad + \frac{L^2}{2}\|\boldsymbol{w}_k - \boldsymbol{w}_{k-1}\|^2 + \langle F(\boldsymbol{w}_{k-1}) - F(\boldsymbol{w}_{k-1}; \xi_{k-1}), \boldsymbol{w}_{k-1} - \boldsymbol{w}\rangle,
\end{aligned} \tag{81}
$$

where $(a)$ is by the fact $ab \leq \frac{a^2}{2} + \frac{b^2}{2}$.

So taking expectation on $\xi_{k-1}$, by Assumption 5, we have: $\forall \boldsymbol{w} \in \mathcal{W}$

$$
\begin{aligned}
&\mathbb{E}_{\xi_{k-1}}[\langle F(\boldsymbol{w}_{k-1}) - F(\boldsymbol{w}_{k-1}; \xi_{k-1}), \boldsymbol{w}_{k-1} - \boldsymbol{w}\rangle] \\
&= \langle \mathbb{E}_{\xi_{k-1}}[F(\boldsymbol{w}_{k-1}) - F(\boldsymbol{w}_{k-1}; \xi_{k-1})], \boldsymbol{w}_{k-1} - \boldsymbol{w}\rangle \\
&= \langle F(\boldsymbol{w}_{k-1}) - F(\boldsymbol{w}_{k-1}), \boldsymbol{w}_{k-1} - \boldsymbol{w}\rangle \\
&= 0.
\end{aligned} \tag{82}
$$

By Assumption 5, we have

$$
\mathbb{E}_{\xi_{k-1}}\Big[\sup_{\boldsymbol{w}\in\mathcal{W}, \|\boldsymbol{w}_k - \boldsymbol{w}\|\leq D} \langle F(\boldsymbol{w}_k), \boldsymbol{w}_k - \boldsymbol{w}\rangle\Big]
$$
$$
\leq \big(1 + \frac{\delta}{\alpha\gamma}\big)LD\,\mathbb{E}_{\xi_{k-1}}[(\|\boldsymbol{w}_k - \boldsymbol{z}_{k-1}\| + \|\boldsymbol{w}_{k-1} - \boldsymbol{z}_{k-1}\|)] + \frac{L^2}{2}\mathbb{E}_{\xi_{k-1}}[\|\boldsymbol{w}_k - \boldsymbol{w}_{k-1}\|^2] + \frac{s^2}{2L^2}.
$$

Lemma 7 is proved.

### D.4 Proof of Theorem 2

*Proof.* Firstly, by the setting $a_k = \frac{\alpha\gamma\sqrt{1+\sigma A_{k-1}}}{L}$ and $A_0 = 0$, $A_k = A_{k-1} + a_k$, we have

- If $\sigma = 0$, then $A_k = \frac{\alpha\gamma k}{L}$.
- If $\sigma > 0$, then $A_k = \left(\frac{\alpha\gamma}{4L}\right)^2 \sigma(k+1)^2$.

Then for both the setting $\sigma = 0$ (*i.e.*, Assumption 3 holds) and $\sigma > 0$ (*i.e.*, Assumption 4 holds), we have

$$\langle F(\boldsymbol{w}_k), \boldsymbol{w}_k - \boldsymbol{w}^* \rangle \geq \frac{\sigma}{\gamma}(V_{\boldsymbol{w}_k - \boldsymbol{w}_0}(\boldsymbol{w}^* - \boldsymbol{w}_0) + V_{\boldsymbol{w}^* - \boldsymbol{w}_0}(\boldsymbol{w}_k - \boldsymbol{w}_0)). \tag{83}$$

So in Lemma 5, let $\boldsymbol{u} = \boldsymbol{w}^*$, we have

$$
\begin{aligned}
0 \quad &\leq \quad \mathbb{E}\Big[\sum_{k=1}^{K} \frac{\sigma a_k}{2}\|\boldsymbol{w}_k - \boldsymbol{w}^*\|^2\Big] \\
&\stackrel{(a)}{\leq} \quad \mathbb{E}\Big[\sum_{k=1}^{K} \frac{\sigma a_k}{\gamma} V_{\boldsymbol{w}^* - \boldsymbol{w}_0}(\boldsymbol{w}_k - \boldsymbol{w}_0)\Big] \\
&\stackrel{(b)}{\leq} \quad \mathbb{E}\Big[\sum_{k=1}^{K} a_k\left(\langle F(\boldsymbol{w}_k), \boldsymbol{w}_k - \boldsymbol{w}^*\rangle - \frac{\sigma}{\gamma}V_{\boldsymbol{w}_k - \boldsymbol{w}_0}(\boldsymbol{w}^* - \boldsymbol{w}_0)\right)\Big] \\
&\stackrel{(c)}{\leq} \quad \mathbb{E}\Big[\sum_{k=1}^{K} E_{2k} + \frac{1}{2\gamma}\|\boldsymbol{w}^* - \boldsymbol{w}_0\|^2\Big] \\
&\stackrel{(d)}{\leq} \quad -\mathbb{E}\Big[4\alpha\sum_{k=1}^{K}(\|\boldsymbol{w}_k - \boldsymbol{z}_{k-1}\|^2 + \|\boldsymbol{w}_{k-1} - \boldsymbol{z}_{k-1}\|^2)\Big] + \frac{8\alpha s^2 K}{L^2} + \frac{1}{2\gamma}\|\boldsymbol{w}^* - \boldsymbol{w}_0\|^2, \tag{84}
\end{aligned}
$$

where $(a)$ is by the $\gamma$-strong convexity Bregman divergence of $V_{\boldsymbol{w}^* - \boldsymbol{w}_0}(\boldsymbol{w}_k - \boldsymbol{w}_0)$, $(b)$ is by the Assumption 3 ($\sigma = 0$) or the Assumption 4 ($\sigma > 0$), $(c)$ is by Lemma 5, and $(d)$ is by Lemma 6.

After a simple arrangement, we have

$$\mathbb{E}\Big[\sum_{k=1}^{K} \frac{1}{K}(\|\boldsymbol{w}_k - \boldsymbol{z}_{k-1}\|^2 + \|\boldsymbol{w}_{k-1} - \boldsymbol{z}_{k-1}\|^2)\Big] \leq \frac{\|\boldsymbol{w}^* - \boldsymbol{w}_0\|^2}{8\alpha K} + \frac{2s^2}{L^2}. \tag{85}$$

Then by randomly picking a $\tilde{k} \in [K]$ with probability distribution $\left\{\frac{a_1}{A_K}, \frac{a_2}{A_K}, \ldots, \frac{a_K}{A_K}\right\}$ and let the output $\tilde{\boldsymbol{w}}_K := \boldsymbol{w}_{\tilde{k}}$, then taking expectation on $\tilde{\boldsymbol{w}}_K$

$$\mathbb{E}_{\tilde{k}}[\|\boldsymbol{w}_{\tilde{k}} - \boldsymbol{z}_{\tilde{k}-1}\|^2 + \|\boldsymbol{w}_{\tilde{k}-1} - \boldsymbol{z}_{\tilde{k}-1}\|^2] \quad = \quad \sum_{k=1}^{K} \frac{a_k}{A_K}(\|\boldsymbol{w}_k - \boldsymbol{z}_{k-1}\|^2 + \|\boldsymbol{w}_{k-1} - \boldsymbol{z}_{k-1}\|^2). \tag{86}$$

So taking expectation on all the history, we have

$$\mathbb{E}[\|\boldsymbol{w}_{\tilde{k}} - \boldsymbol{z}_{\tilde{k}-1}\|^2 + \|\boldsymbol{w}_{\tilde{k}-1} - \boldsymbol{z}_{\tilde{k}-1}\|^2] \leq \frac{\|\boldsymbol{w}^* - \boldsymbol{w}_0\|^2}{8\alpha K} + \frac{2s^2}{L^2}. \tag{87}$$

Then taking expectation on all the history, we have

$$
\begin{aligned}
&\mathbb{E}[\|\boldsymbol{w}_{\tilde{k}} - \boldsymbol{z}_{\tilde{k}-1}\| + \|\boldsymbol{w}_{\tilde{k}-1} - \boldsymbol{z}_{\tilde{k}-1}\|] \\
&\stackrel{(a)}{\leq} \quad (\mathbb{E}[(\|\boldsymbol{w}_{\tilde{k}} - \boldsymbol{z}_{\tilde{k}-1}\| + \|\boldsymbol{w}_{\tilde{k}-1} - \boldsymbol{z}_{\tilde{k}-1}\|)^2])^{1/2} \\
&\stackrel{(b)}{\leq} \quad (\mathbb{E}[2(\|\boldsymbol{w}_{\tilde{k}} - \boldsymbol{z}_{\tilde{k}-1}\|^2 + \|\boldsymbol{w}_{\tilde{k}-1} - \boldsymbol{z}_{\tilde{k}-1}\|^2)])^{1/2} \\
&\stackrel{(c)}{\leq} \quad \sqrt{2}\sqrt{\frac{\|\boldsymbol{w}^* - \boldsymbol{w}_0\|^2}{8\alpha K} + \frac{2s^2}{L^2}}, \tag{88}
\end{aligned}
$$

where $(a)$ is by the Jensen inequality, $(b)$ is by the fact that $(a+b)^2 \leq 2(a^2 + b^2)$ and $(c)$ is by (87).

$$
\mathbb{E}\Big[\sup_{\boldsymbol{w}\in\mathcal{W}, \|\boldsymbol{w}_{\tilde{k}}-\boldsymbol{w}\|\leq D}\langle F(\boldsymbol{w}_{\tilde{k}}), \boldsymbol{w}_{\tilde{k}}-\boldsymbol{w}\rangle\Big]
$$

$$
\overset{(a)}{\leq} \mathbb{E}\Big[\Big(1+\frac{\delta}{\alpha\gamma}\Big)LD(\|\boldsymbol{w}_{\tilde{k}}-\boldsymbol{z}_{\tilde{k}-1}\|+\|\boldsymbol{w}_{\tilde{k}-1}-\boldsymbol{z}_{\tilde{k}-1}\|)
$$

$$
+\frac{L^2}{2}\|\boldsymbol{w}_{\tilde{k}}-\boldsymbol{w}_{\tilde{k}-1}\|^2+\frac{1}{2L^2}\|F(\boldsymbol{w}_{\tilde{k}-1})-F(\boldsymbol{w}_{\tilde{k}-1};\xi_{\tilde{k}-1})\|_*^2\Big]
$$

$$
\overset{(b)}{\leq} \mathbb{E}\Big[\Big(1+\frac{\delta}{\alpha\gamma}\Big)LD(\|\boldsymbol{w}_{\tilde{k}}-\boldsymbol{z}_{\tilde{k}-1}\|+\|\boldsymbol{w}_{\tilde{k}-1}-\boldsymbol{z}_{\tilde{k}-1}\|)+\frac{L^2}{2}(\|\boldsymbol{w}_{\tilde{k}}-\boldsymbol{z}_{\tilde{k}-1}\|+\|\boldsymbol{w}_{\tilde{k}-1}-\boldsymbol{z}_{\tilde{k}-1}\|)^2\Big]
$$

$$
+\frac{s^2}{2L^2}
$$

$$
\overset{(c)}{\leq} \mathbb{E}\Big[\Big(1+\frac{\delta}{\alpha\gamma}\Big)LD(\|\boldsymbol{w}_{\tilde{k}}-\boldsymbol{z}_{\tilde{k}-1}\|+\|\boldsymbol{w}_{\tilde{k}-1}-\boldsymbol{z}_{\tilde{k}-1}\|)+L^2(\|\boldsymbol{w}_{\tilde{k}}-\boldsymbol{z}_{\tilde{k}-1}\|+\|\boldsymbol{w}_{\tilde{k}-1}-\boldsymbol{z}_{\tilde{k}-1}\|)^2\Big]
$$

$$
+\frac{s^2}{2L^2}
$$

$$
\overset{(d)}{\leq} \sqrt{2}\Big(1+\frac{\delta}{\alpha\gamma}\Big)LD\sqrt{\frac{\|\boldsymbol{w}^*-\boldsymbol{w}_0\|^2}{8\alpha K}+\frac{2s^2}{L^2}}+L^2\Big(\frac{\|\boldsymbol{w}^*-\boldsymbol{w}_0\|^2}{8\alpha K}+\frac{2s^2}{L^2}\Big)+\frac{s^2}{2L^2}, \tag{89}
$$

where $(a)$ is by Lemma 7, $(b)$ is by the triangle inequality of $\|\cdot\|$ and Assumption 5, $(c)$ is by the triangle inequality of $\|\cdot\|$, $(d)$ is by (87) and (88).

For $\sigma>0$, by (84) and (85), we have

$$
\mathbb{E}\Big[\sum_{k=1}^{K}\frac{a_k\sigma}{2}\|\boldsymbol{w}_k-\boldsymbol{w}^*\|^2\Big]
$$

$$
\leq -\mathbb{E}\Big[4\alpha\sum_{k=1}^{K}(\|\boldsymbol{w}_k-\boldsymbol{z}_{k-1}\|^2+\|\boldsymbol{w}_{k-1}-\boldsymbol{z}_{k-1}\|^2)\Big]+\frac{8\alpha s^2 K}{L^2}+\frac{1}{2\gamma}\|\boldsymbol{w}^*-\boldsymbol{w}_0\|^2
$$

$$
\leq \frac{8\alpha s^2 K}{L^2}+\frac{1}{2\gamma}\|\boldsymbol{w}^*-\boldsymbol{w}_0\|^2.
$$

So by the definition of $\tilde{\boldsymbol{w}}_K$, taking expectation on the randomness of all the history, we have

$$
\mathbb{E}[\|\boldsymbol{w}_{\tilde{k}}-\boldsymbol{w}^*\|^2]
$$

$$
\leq \mathbb{E}\Big[\sum_{k=1}^{K}\frac{a_k}{A_K}\|\boldsymbol{w}_k-\boldsymbol{w}^*\|^2\Big]
$$

$$
\leq \frac{2}{\sigma A_K}\Big(\frac{8\alpha s^2 K}{L^2}+\frac{1}{2\gamma}\|\boldsymbol{w}^*-\boldsymbol{w}_0\|^2\Big)
$$

$$
\leq \frac{32L^2}{\sigma^2(\alpha\gamma)^2(K+1)^2}\Big(\frac{8\alpha s^2 K}{L^2}+\frac{1}{2\gamma}\|\boldsymbol{w}^*-\boldsymbol{w}_0\|^2\Big).
$$

Theorem 2 is proved. ∎