[Reviews · NeurIPS 2020]

Review 1

Summary and Contributions: Variational inequality (VI) has many important application, e.g., min-max optimization, game theory, etc. VI with monotone opertors has been well studied. This paper considers VI with non-monotone operators. The authors developed an optimistic dual extrapolation (ODE) method, and show that the method converges and characterize the convergence rate under two non-monotone settings. In addition, the method is generalized to stochastic setting.

Strengths: This paper propose an ODE method for VI with non-monotone operators. It is shown that ODE converges to a strong solution under weaker assumptions. The rate of convergence for ODE is characterized and matches the best state-of-the-art methods. As byproduct, the authors proposed the first near optimal convergence guarantee. The authors also extended ODE to stochastic setting (SODE). The results of this paper are interesting, and made some meaningful contributions for gradient-based methods for variational inequalities. I would like to raise my score.

Weaknesses: (1) The parameters on input of the algorithm are unclear. How to choose them in practice? (2) No simulation is provided to support the theoretical results.

Correctness: The result looks right. However, I didn't check the proof.

Clarity: Yes. A minor advice, ODE is somewhat misleading, because it is usually referred to as "ordinary differential equation", consider an alternative abbrv please.

Relation to Prior Work: Yes

Reproducibility: Yes

Additional Feedback:


Review 2

Summary and Contributions: This paper proposes a single step version of Nesterov's dual extrapolation method. The authors prove last iterate convergence rates under weak monotonicity assumption (precisely that a weak solution of the VIP exists) The authors prove convergence rates for the deterministic and stochastic cases. ==== AFTER REBUTTAL ==== I have read the authors' rebuttal as well as the other reviews, and I would like to point out that the inconsistencies between the appendix and the main text pointed out by the other reviewers must be fixed. However, I still think the results in this paper are interesting, and I appreciated the authors' response. Theorem 2 is novel and to the interest of the community. More precisely the \|x_0 -x^*\| dependence in the bound of Thm 2 is novel and very interesting. I provide additional feedback below that needs to be addressed in the camera-ready version One thing that should be more discussed is what you called in your rebuttal 'the main point of Section 4'. For your algorithm, it seems that your result indicates that "one cannot obtain provable convergence rates by only decreasing the step size, whereas a large batch size is necessary." However, as I mentioned in my review, the standard algorithm for stochastic VIP can reduce the effect of noise with a smaller step-size (see, for instance, Thm 2, 3 & 4 in [12] of your paper). Is this discrepancy btw SODE and standard method (e.g., stochastic gradient or stochastic extragradient) due to an intrinsic difference in the optimization methods, or is it just an artifact of a different proof technique? (Same question with \|x_0 -x^*\| appearing in your bounds and not in the bounds provided in Thm 2, 3 & 4 in [12]: is this \|x_0 -x^*\| dependence in the bound of your Thm 2 characteristic of your method or just due to different proof techniques?) That is also why I find the claim "Thus our result partly validates the good empirical performance of large batch size in the training of GAN" needs to be contrasted with 1) the fact that standard methods such as stochastic extragradient and stochastic gradient do not require an increasing batch size to reduce the noise. 2) in [3], large batch sizes in GANs have been used with the stochastic gradient method... not SODE. Also, I think the new proposed title, "Optimistic Dual Extrapolation for a Class of Nonmonotone Variational Inequalities," is still a bit misleading. The assumption made is very similar to the coherence assumption see Mertikopoulos et al. [2019] or Stochastic Mirror Descent in Variationally Coherent Optimization Problems Zhou et al. A more precise title could be 'Optimistic Dual Extrapolation for coherent (Nonmonotone) Variational Inequalities' Reference: Mertikopoulos et al. 2019: https://openreview.net/pdf?id=Bkg8jjC9KQ Minor comment: ODE usually stands for Ordinary differential equations. It may be a good idea to find another acronym.

Strengths: This paper is well written. Last iterate convergence is a challenging question in optimization for VIPs. The results are strong and interesting. I also find that there is a strength is this work that is not emphasized by the authors: the standard results for VIP do not contain \|x_0 -x^*\| in their convergence bound (see [Thm 2, 1] and [Thm.1, 2]) but only D. That is a significant improvement when D is large, and the initialization x_0 is close to x^*. To my knowledge, such dependence on in the constant is new. Are the authors aware of any previous work providing such dependence? Otherwise, it could be a contribution to the paper. \|x_0 -x^*\| My question is the following: Do you think similar proof techniques could be applied to extragradient (for instance) to replace D by \|x_0 -x^*\| in [Thm 2, 1] or do you think it is a particular property of your method? However, note that I have concerns that should be addressed. (see correctness section) [1] Nesterov, Yurii. "Dual Extrapolation and its Applications for Solving Variational Inequalities and Related Problems'." Available at SSRN 988671 (2003). [2] Juditsky, Anatoli, Arkadi Nemirovski, and Claire Tauvel. "Solving variational inequalities with stochastic mirror-prox algorithm." Stochastic Systems 1.1 (2011): 17-58.

Weaknesses: One major weakness of this work is that the author did not run any experiments to validate their theoretical findings (namely, last convergence iterate in the deterministic and stochastic case). Mainly, if Theorem 2 is valid, experiments with the proposed method should exhibit last iterate convergence in the stochastic setting, which is not the case for extragradient [Thm1, 1]. Such an experimental result would be a strong argument in favor of optimistic methods versus extragradient. [1] Chavdarova, Tatjana, et al. "Reducing noise in GAN training with variance reduced extragradient." Advances in Neural Information Processing Systems. 2019.

Correctness: I have mainly two questions: the first one regards your restricted strong merit function: It is not exactly the one proposed by Nesterov [30]. In (10) you should replace \|w- \tilde w_K\| by \|w - \bar w\| where \bar w is a point that *does not* depend on the current iterate. (see Eq (2.3) in [1]). Can you comment on the difference between these two merit functions? It seems that (4) is *not* a merit function in general for a fixed R since the solution w^* can potentially not belong to the set of w such that \|w-\tilde w_K\| \leq D. Can you comment on that? I am a bit confused by the missing dependence on the step-size in Theorem 2. Usually, the variance of the noise (in you case s^2) is multiplied by the step-size (in you case \alpha). Usually, the convergence rate are of this form O(1/(\alpha K) + \alpha s^2) see for instance [Thm.1, 2]. there is then a tradeoff: with a small step-size you then can diminish the impact of the noise but increase the constant in front of O(1/(\alpha K)). Why doesn’t this tradeoff appear in Theorem 2? [1] Nesterov, Yurii. "Dual Extrapolation and its Applications for Solving Variational Inequalities and Related Problems'." Available at SSRN 988671 (2003). [2] Juditsky, Anatoli, Arkadi Nemirovski, and Claire Tauvel. "Solving variational inequalities with stochastic mirror-prox algorithm." Stochastic Systems 1.1 (2011): 17-58.

Clarity: This paper is well written. However, I find the mention of ‘non-monotone’ in the title and the contribution is a bit of an overclaim. The general non-monotone case is not handled (which is fine but the title may be misleading). The assumption made is closer to a ‘weakened’ monotonicity assumption than a general non-monotone setting.

Relation to Prior Work: The prior work is discussed well.

Reproducibility: Yes

Additional Feedback: L35: 'it is well known' If it is well known can you put a reference? Footnote 1 P2: the sentence is a bit confusing. L 138: I would be clearer to say “by (6)’ instead of ‘Obviously’ Page 10: (43) and above it should be w instead of z on the RHS. ==== AFTER REBUTTAL ==== L290 in the appendix is broken.


Review 3

Summary and Contributions: %%%%%%%%%%%% Update after rebuttal %%%%%%%%%%%% I would like to thank the authors for their reply. However, I still believe the use of the term last-iterate convergence is very confusing and should be revised (I perfectly understand the authors' explanation and their contribution, but I would just like to point out currently this result is not so well presented). Moreover, Corollary 1 in the main text is significantly different from Corollary 1 in the supplement (while none of the two seem to be the correct statement according to the proof) and this should also be fixed. While I will not lower my score, I suggest the authors put some more efforts to improve their presentation (whether the paper finally gets accepted or not). --------------------------------------------------------------------------------------------------- The paper contributes towards developing more efficient methods for variational inequalities by designing a single-call version of Nesterov's dual extrapolation algorithm. This new algorithm is named optimistic dual extrapolation and is shown to enjoy several theoretical guarantees under various assumptions. More precisely, a convergence rate of (1/ε^2) in terms of the restricted primal gap is proved when a weak solution exists and the linear convergence is proved when a strong weak solution exists (a slightly weaker assumption than strong variational stability). For monotone functions, O(1/ε log(1/ε)) convergence guarantee in terms of restricted primal gap is derived. Similar results are established for the stochastic setting by using large batch sizes to reduce variance if necessary.

Strengths: The paper makes several worthy contribution towards the design and analysis of gradient-based methods for variational inequalities. First, although the extension of dual extrapolation to the single-call setup is straightforward, I am personally not aware of any existing literature that formally presents this method. As far as I know, the most similar algorithm is probably optimistic FTRL which only differs in the construction of w_k (with the σ=0 setup). Moreover, for this algorithm, new non-asymptotic convergence guarantees are derived, of particularly interest is probably a linear convergence bound on the primal gap function when a strongly weak solution exists.

Weaknesses: Here I make several comments on Corollary 1 and its surrounding paragraphs; please refer to the following cases for other points. For Corollary 1 and its surrounding paragraphs: In my opinion, the authors should properly define what they call by last-iterate convergence. In general, an algorithm is said to exhibit last-iterate convergence if the generated sequence converges to a solution. For a non-asymptotic type of result, it is generally shown that at every iteration k some convergence measure of w_k is bounded. Nonetheless, Corollary 1 as stated in the appendix requires to set the accuracy ε before running the algorithm and the iterate is only guaranteed to converge to an ε-approximated strong solution. Therefore, this seems to be different from what people usually expect and should be clearly explained. Moreover, as for the asymptotic convergence, it was already shown in the original paper of Popov [1] for a variant of single-call methods in the monotone setting and in various papers for extragradient methods under different non-monotone assumptions (pseudo-monotone, variational stability, etc.). Therefore, lines 200-204 would only make sense if the authors were talking about the "last-iterate convergence rate". [1] Popov, L. D. (1980). A modification of the Arrow-Hurwicz method for search of saddle points. Mathematical notes of the Academy of Sciences of the USSR, 28(5), 845-848

Correctness: The claims of the paper look reasonable and are in line with existing results for similar methods in the literature.

Clarity: In my mind, the writing and the organization of the paper can be improved in several ways. First, the authors could describe in more detail the background of the studied problem and the involved notions. Examples are the derivation of the VI problem, the properties of different gap functions and a comparison of the assumptions appearing in the literature. More importantly, I think the discussion on the natural residual should be mentioned earlier, probably already in the introduction. In effect, in Table 2 the authors implicitly use Proposition 3 in the supplement to derive the rate from (Iusem et al. 2017) while this is not explained. In addition, for extragradient we can easily derive a bound on the natural residual which translates to a bound on the primal gap, while for optimistic methods the authors rely on a similar but slightly different quantity. This seems to be key step to the proof and can also be discussed in the main text. Finally, the authors could explain why they choose to study dual extrapolation type of algorithm instead of mirror-prox type methods if there is any specific reason behind this. Typo: z0 instead of v0 in the two pseudo-codes

Relation to Prior Work: As mentioned earlier, there should be more discussion on various convergence measures used in the VI literature. In particular, in a recent paper [1] the result of (Iusem et al. 2017) is extended to optimistic gradient descent under the exact assumption that a weak solution exists. Its results is very similar to the one presented in this paper and mainly differs in two points: (i) the projections are different (agile versus lazy type of projection), (ii) the convergence measures are different (primal gap versus natural residual). Besides this, from line 96 to 99 the authors claim that (Golowich et al. 2020) proves some result about optimistic method in non-monotone setting, which is not true (since that paper concerns extragradient for monotone VIs). I also fail to see how optimistic dual extrapolation presented here is related to the regularized dual averaging method proposed in (Xiao 2010) since no composite term appears in the algorithm, which makes the use of L1 regularization impossible (Remark 2). [1] Liu, M., Mroueh, Y., Ross, J., Zhang, W., Cui, X., Das, P., & Yang, T. (2019, September). Towards Better Understanding of Adaptive Gradient Algorithms in Generative Adversarial Nets. In International Conference on Learning Representations.

Reproducibility: Yes

Additional Feedback:


Review 4

Summary and Contributions: This paper studies a single-call extragradient method for solving a class of non-monotone variational inequalities. The algorithm is a hybrid version of dual averaging and the extragradient method. Both stochastic and deterministic VIs under weak monotonicity assumptions are studied. Emphasis is put on convergence of the last iterate, and optimal complexity results are obtained.

Strengths: The paper provides an innovative hybrid single-call extragradient method. Convergence of the last iterate is proven and as well as non-asymptotic optimal iteration complexity. This is an interesting result but somewhat expected. What I personally find most intriguing is the design of the algorithm. Including the gradient of the Bregman divergence into the dual update step as a regularizer is a nice innovation, and resembles classical Tikhonov regularization ideas. This could lead to interesting new algorithmic developments.

Weaknesses: The main paper needs a serious revision in terms of writing. There seems to be a discrepancy between the content in the supplementary materials and the main paper. In particular, the following points should be taken into consideration: ▪ I don’t understand why only strongly convex norms can be used as distance generating kernels. It seems that you insists on kernels which are not steep at the boundary. I don’t know exactly why, so please provide some more details here. ▪ The notation should be improved. Usually we call distance generating kernels by omega or h, so please choose one of them also in the paper. Write the gradient of the Bregman divergence in terms of the function and not in terms of the distance. This is unpleasant to read. ▪ v_{0} in line 7 of Algorithm 1 and Algorithm 2 is not defined ▪ In know that in ML the word „optimistic“ is common. I never understood what this means. Maybe you can add a short explanation. Corollary 2: Missing element symbol.

Correctness: As far as I can say the results seem to be correct.

Clarity: This is the major weakness of the paper and the reason why I have to lower my overall score. The paper needs a very serious revision in order to be acceptable.

Relation to Prior Work: The references mentioned are adequate.

Reproducibility: Yes

Additional Feedback: ====Update after the rebuttal==== I think this paper contains some very interesting parts. However, the submission has sever deficits in terms of the writing, and I am not convinced that the paper will be ready for publication after a short revision. For this reason I lower my final score.

[Author Response · NeurIPS 2020]

**R1:** Thanks for your positive evaluation! There are four parameters in both Algorithm 1 and Algorithm 2: $L, \gamma, \delta$ and $\sigma$. $L$ is the Lipschitz constant of the operator, which is tuned in implementation or alternatively handled by adopting an extra parameter-free strategy. $\gamma, \delta$ are parameters of the strongly convex norm square $\frac{1}{2}\|\cdot\|^2$, which can be easily verified in practice. In line 133-134, we have shown the values of $\gamma$ and $\delta$ for the $p$-norm $\frac{1}{2}\|\cdot\|_p^2$. $\sigma$ is the constant in Assumption 3 ($\sigma = 0$) or Assumption 4 ($\sigma > 0$). In practice, the nonzero $\sigma$ is often obtained by an explicit $\ell_2$-norm regularization, so it can also be verified effectively. In camera-ready, we will include a paragraph to show the choice of these parameters. Regarding the lack of simulations, we will add numerical experiments to compare our algorithm with existing ones in camera-ready. Finally, the abbrv ODE is somewhat misleading indeed. We will use OptDE instead.

**R2:** Thanks for your positive feedback! Thanks also for pointing out the extra strength that we were not paying attention to. By Dang&Lan 2015 [8], it seems that such a strength also exists for extragradient-type methods. We believe it is a natural by-product from proving approximate strong solution guarantees, while both results in [Thm 2, 1] and [Thm.1, 2] are in terms of approximate weak solution guarantees. On the simulations front, per your suggestion, we will do experiments to validate the behavior of our algorithms, particularly for the last-iterate convergence.

Thanks for your insightful observation in terms of the definition of restricted strong merit function! Our definition is not an exact analog of Nesterov [30] indeed. However, as shown in page 329 of [30], the restricted merit function will only be informative when $D$ satisfies $D \geq \|\boldsymbol{w}^* - \bar{\boldsymbol{w}}\|$, where $\boldsymbol{w}^*$ is the solution of a monotone problem. This is because, by Lemma 1 of [30], only under the condition $D \geq \|\boldsymbol{w}^* - \bar{\boldsymbol{w}}\|$ do we get the following: the solution that makes the restricted merit function 0 is the solution of the underlying monotone problem. Consequently, since only a large $D$ is informative and the value of $D$ only appears in the theoretical guarantee, we do not need to worry about what will happen in the case of small $D$: we can just pick a large enough $D$ to make $\boldsymbol{w}^*$ contained in the set. In camera-ready, we will explicitly discuss this point.

Additionally, you totally got the main point in Section 4! For minimization problems, we can reduce the effect with a small step size (i.e., learning rate). However, in the nonmonotone setting (Assumption 3), possibly due to the lack of a Lyapunov function and the inability of performing averaging *simultaneously*, "one cannot obtain provable convergence rates by only decreasing the step size, whereas a large batch size is necessary"(line 265-268). We believe such a fact partly validates why we must use a large batch size in the training of GAN. Finally, we will change the title to "Optimistic Dual Extrapolation for a Class of Nonmonotone Variational Inequalities" to avoid the possibility of overclaim. Thanks for catching all the typos: consider all of them fixed.

**R3:** Thanks for your positive evaluation! In this paper, we are mainly concerned with computational complexity in finding an $\epsilon$-accurate solution. If we hope to guarantee the convergence in the strict sense of last-iterate, the best possible rate will be $O(1/\epsilon^2)$ for extragradient (EG) even in the monotone setting [13]. Meanwhile, optimistic methods can be viewed as approximations of EG and the nonmonotone setting includes the monotone one as an instance. Thus we can not expect a better rate than $O(1/\epsilon^2)$ rate in the strict sense of last-iterate for optimistic methods in the nonmonotone setting. To avoid the $O(1/\epsilon^2)$ barrier, we relax the concept of last iterate convergence as follows: we only guarantee the convergence rate when the iterate $k \geq O(1/\epsilon)$. For the beginning $k \leq O(1/\epsilon)$ iterations, the last iterate may not necessarily converge. Additionally, going beyond "asymptotic convergence" (which only characterizes qualitative convergence when the number of iteration tends to $\infty$), we provide explicit finite-time convergence rates. Furthermore, in optimization, it is standard to treat the accuracy parameter $\epsilon$ as an input of an algorithm. The regularization trick in this paper depends on the specification of $\epsilon$ beforehand. Of course, developing algorithms that are agnostic to knowing $\epsilon$ is interesting (and significantly but beyond the scope of this paper) and we leave it for future research.

Thank you for your detailed writing suggestions! Following them, we will discuss more about the derivation of the VI problem, the properties of different gap functions and a comparison of the assumptions appearing in the literature. Meanwhile, we will move the discussion of natural residual earlier to make the result about Iusem et al. 2017 in Table 2 more clear. Thanks also for pointing out the relevant ICLR reference! In addition to the difference you mentioned, we also study the setting where *a strongly weak solution exists* while they did not. The results of this setting are significant as they allow us to obtain near-optimal approximate strong solution guarantees for the monotone setting. As mentioned in Remark 2, we consider the dual extrapolation approach because we can give a unified convergence analysis under Assumptions 3 and 4 using estimation sequence. Meanwhile, if there exists a regularizer, the lazy update can exploit the structure of regularizer better. For simplicity, we did not consider a composite term in the algorithm. However, it can be handled in the same way as the constrained set, if a certain efficient proximal operator exists for the composite term.

**R4:** Thanks for your positive feedback! We will reorganize the writing according to you and **R3**. We must use strongly convex norms, where the strong convexity is used to cancel certain errors in the convergence analysis; additionally, it also makes the solution of subproblems unique. Following your suggestion, we will define the distance generating kernels by $h$ and make the gradient in terms of the function. We will correct all the typos you pointed out. The term "optimistic" is coined by Rakhlin and Sridharan [35] in the online learning context, which then has been used in a confusing way in the existing literature indeed. In our context, we do not "conservatively" compute a new gradient but instead reuse the computed past gradients for the extrapolation step, which is thus an "optimistic" procedure.

[Meta-Review · NeurIPS 2020]

Originally, all reviewers recommended weak accept (6). This paper presents a single step version of Nesterov's dual extrapolation method (and a stochastic variant), with guarantees for coherent (nonmonotone) variational inequalities (see R2 Q1). The reviewers underlined the algorithmic contribution, as well as the interesting theoretical contribution of this work (though there is no empirical study of its performance). However, several concerns were expressed about the quality of the writing, especially the disconnect between the appendix and the main paper. The reviewers carefully considered the rebuttal and discussed the work. After the rebuttal, R4 decided to downgrade their score to weak reject (5), mainly because they thought the write-up needed a major revision. On the other hand, R2 in discussion argued that the paper should be accepted given its nice contributions, despite some issues with the presentation, and upgraded their rating to a 7.. They were hesitating between a 6 and 7 for their score, and thought that the authors could revise their paper fine for the camera ready version to address the reviewers concerns. All reviewers agreed in discussion that the paper made nice algorithmic and theoretical contribution. The AC also considered that this paper seems to be a significant improvement over its ICML submission, where it was already close to be accepted. The authors improved the novelty of the work and the theoretical contributions by proposing a single step version of the algorithm (instead of the traditional two steps like in their ICML submission), yielding novel proof techniques. Given the above, the AC agrees with R2 and does not think another round of reviewing is necessary for this paper, and recommends acceptance. The authors should carefully consider the reviews and implement the asked changes in the camera ready version. Among others, the title change from R2; important clarifications from all reviewers; adding the simulation results as proposed by R2; and careful re-writing of appendix to be consistent with main paper.